# Negative feedback couples Hippo pathway activation with Kibra degradation independent of Yorkie-mediated transcription

Sherzod A Tokamov[1,2], Ting Su[1†], Anne Ullyot[1], Richard G Fehon[1,2]*

[1]Department of Molecular Genetics and Cell Biology, The University of Chicago, Chicago, United States; [2]Committee on Development, Regeneration and Stem Cell Biology, The University of Chicago, Chicago, United States

**Abstract** The Hippo (Hpo) pathway regulates tissue growth in many animals. Multiple upstream components promote Hpo pathway activity, but the organization of these different inputs, the degree of crosstalk between them, and whether they are regulated in a distinct manner is not well understood. Kibra (Kib) activates the Hpo pathway by recruiting the core Hpo kinase cassette to the apical cortex. Here, we show that the Hpo pathway downregulates *Drosophila* Kib levels independently of Yorkie-mediated transcription. We find that Hpo signaling complex formation promotes Kib degradation via SCF$^{Slimb}$-mediated ubiquitination, that this effect requires Merlin, Salvador, Hpo, and Warts, and that this mechanism functions independently of other upstream Hpo pathway activators. Moreover, Kib degradation appears patterned by differences in mechanical tension across the wing. We propose that Kib degradation mediated by Hpo pathway components and regulated by cytoskeletal tension serves to control Kib-driven Hpo pathway activation and ensure optimally scaled and patterned tissue growth.

*For correspondence:
rfehon@uchicago.edu

Present address: †Department of Physiology, UT Southwestern Medical Center, Dallas, United States

Competing interests: The authors declare that no competing interests exist.

## Introduction

How organs achieve and maintain optimal size is a fundamental question in developmental biology. The Hippo (Hpo) signaling pathway is an evolutionarily conserved inhibitor of tissue growth that was first identified in *Drosophila* in somatic mosaic screens for tumor-suppressor genes (*Xu et al., 1995*; *Tapon et al., 2002*; *Harvey et al., 2003*; *Wu et al., 2003*). Central to the Hpo pathway activity is a kinase cassette that includes serine/threonine kinases Tao-1, Hpo, and Warts (Wts), as well as two scaffolding proteins Salvador (Sav) and M̲ob a̲s t̲umor s̲uppressor (Mats). Activation of the Hpo pathway results in a kinase cascade that culminates in the phosphorylation of a transcriptional co-activator Yorkie (Yki) by Wts, which inhibits Yki nuclear accumulation. Conversely, inactivation of the Hpo pathway allows Yki to translocate into the nucleus where, together with its DNA-binding partners such as Scalloped (Sd), it promotes transcription of pro-growth genes. As a result, inactivation of the Hpo pathway is characterized by excessive tissue growth. Mutations that disrupt Hpo pathway activity can lead to various human disorders including benign tumors and carcinomas (*Zheng and Pan, 2019*).

A distinct feature of the Hpo pathway is the remarkably complex organization of its upstream regulatory modules (*Fulford et al., 2018*). The core Hpo kinase cascade is regulated from the cell cortex by multiple upstream components, including Fat (Ft), Dachsous (Ds), Echinoid (Ed), Expanded (Ex), Crumbs (Crb), Kibra (Kib), and Merlin (Mer). Broadly speaking, these components localize either exclusively junctionally (a term that we use to include both the adherens junctions and the marginal zone; *Tepass, 2012*) or both junctionally and at the apical medial cortex (*Su et al., 2017*). Ft and Ds

are protocadherins that promote Hpo pathway activity from the junctions by restricting the activity of Dachs, an atypical myosin that inhibits Wts (*Bennett and Harvey, 2006*; *Cho et al., 2006*; *Mao, 2006*; *Matakatsu and Blair, 2012*; *Vrabioiu and Struhl, 2015*). Ed is a cell–cell adhesion protein that binds and stabilizes Sav at the junctional cortex, thereby enabling Sav to promote Hpo pathway activity (*Yue et al., 2012*). Ex is a FERM-domain protein that also localizes at the junctional cortex where it binds to the transmembrane protein Crb and activates the Hpo pathway by recruiting the core kinase cassette (*Hamaratoglu et al., 2006*; *Ling et al., 2010*; *Robinson et al., 2010*; *Sun et al., 2015*). The WW-domain protein Kib and FERM-domain protein Mer localize both at the junctional and apical medial cortex and promote Hpo pathway activity by recruiting the core kinase cassette independently of Ex (*Yu et al., 2010*; *Baumgartner et al., 2010*; *Genevet et al., 2010*; *Hamaratoglu et al., 2006*; *Su et al., 2017*). The existence of multiple upstream regulatory modules that converge to control the activity of a single downstream effector, Yki, raises a question of whether and how these parallel inputs are regulated and to what extent they are distinct from one another.

One way that cells modulate signaling output is by controlling the levels of signaling components. Within the Hpo pathway, transcription of Ex, Kib, and Mer is positively regulated by Yki activity in a negative feedback loop (*Hamaratoglu et al., 2006*; *Genevet et al., 2010*; *Yee et al., 2019*). Multiple Hpo pathway components are also regulated post-translationally. For example, Crb promotes Ex ubiquitination via Skip/Cullin/F-box$^{Slimb}$ (SCF$^{Slimb}$) E3 ubiquitin ligase complex, which leads to Ex degradation (*Ribeiro et al., 2014*; *Fulford et al., 2019*). Similarly, Ds and Dachs levels are downregulated by the SCF$^{Fbxl-7}$ E3 ubiquitin ligase, and Dachs stability is also influenced by an E3 ubiquitin ligase called Early girl (*Bosch et al., 2014*; *Rodrigues-Campos and Thompson, 2014*; *Misra and Irvine, 2019*). Sav stability is also inhibited by the HECT (Homologous to the E6-AP Carboxyl Terminus) ubiquitin ligase Herc4 (*Aerne et al., 2015*). These studies underscore the importance of post-translational regulation of Hpo pathway components and suggest that individual signaling branches of the Hpo pathway might be regulated in a distinct manner from one another.

In this study, we reveal that the Hpo pathway negatively regulates Kib levels via post-translational negative feedback. We show that the regulation of Kib levels by the Hpo pathway is independent of Yki- and Sd-mediated transcriptional output and is instead mediated by SCF$^{Slimb}$. We find that this mechanism operates independently of other upstream inputs, such as Ex/Crb or Ft/Ds, and requires Kib-mediated complex formation. Intriguingly, our data suggest that Kib degradation is regulated by mechanical tension across the wing imaginal tissue. We propose a model in which Kib-mediated Hpo pathway complex formation results in Kib degradation in isolation from other upstream inputs, thereby forming a tightly compartmentalized negative feedback loop. Such feedback may function as a homeostatic mechanism to tightly control signaling output specifically downstream of Kib and ensure proper tissue growth during development.

## Results

### Transcriptional feedback is insufficient to explain the increase in Kib abundance upon pathway inactivation

A notable feature of the Hpo pathway is that its upstream components Kib, Ex, and Mer are upregulated by Yki transcriptional activity in a negative feedback loop (*Hamaratoglu et al., 2006*; *Genevet et al., 2010*; *Yee et al., 2019*). In particular, Kib levels were previously shown to be significantly elevated in double-mutant *Mer*; *ex* somatic mosaic clones, consistent with the transcriptional feedback regulation of *kibra* by Yki (*Genevet et al., 2010*). However, when we examined endogenous Kib tagged with the green fluorescent protein (Kib::GFP) in live wing imaginal discs containing either *Mer* or *ex* mutant clones individually, we found that Kib abundance was significantly higher in *Mer* mutant clones than in *ex* mutant clones (*Figure 1A–C*). These results suggest that loss of Mer has a greater effect on Yki transcriptional activity than loss of Ex, which has not been reported previously.

To directly assess the relative contribution of Mer and Ex to Yki activity, we examined the nuclear localization of endogenously expressed Yki-YFP, a biosensor for Yki activity (*Su et al., 2017*; *Xu et al., 2018*). In sharp contrast to what we observed with Kib levels, Yki strongly accumulated in the nuclei of *ex* mutant clones, whereas Yki was mostly cytoplasmic and indistinguishable from wild-

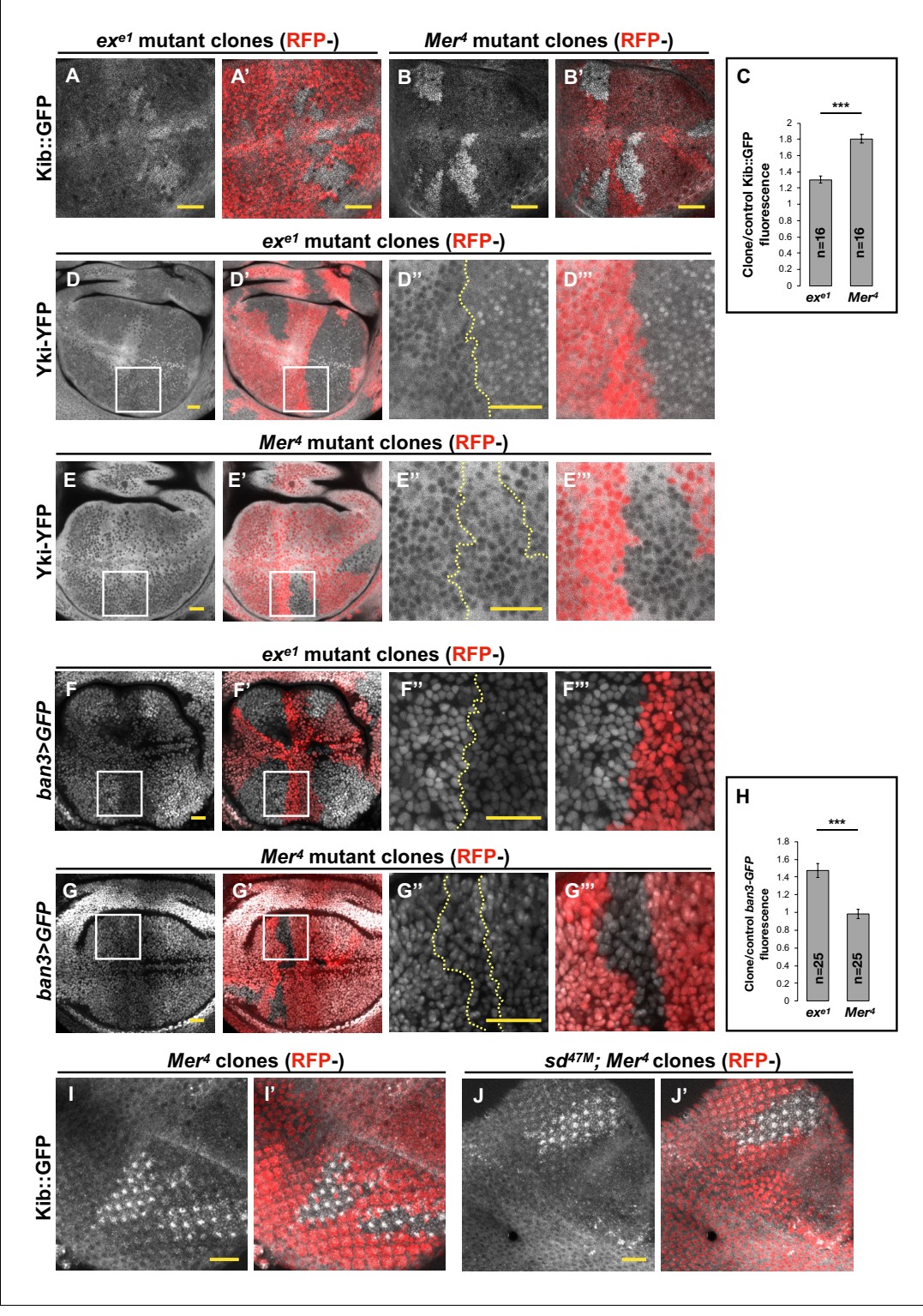

**Figure 1.** Transcriptional feedback alone does not explain Kibra (Kib) upregulation in *Mer* clones. (**A–G'''**) All tissues shown are living late third instar wing imaginal discs expressing the indicated fluorescent proteins. (**A–C**) Endogenous Kib::GFP in *ex* (**A and A'**) or *Mer* (**B and B'**) somatic mosaic clones (indicated by loss of RFP). Loss of Mer leads to a greater increase in Kib levels than loss of Ex. Quantification is shown in (**C**). (**D–E'''**) Endogenously expressed Yorkie (Yki)-YFP is strongly nuclear in *ex* mutant clones (**D–D'''**) but is mostly cytoplasmic in *Mer* mutant clones (**E–E'''**). (**F–H**) Expression *ban3>GFP*, a reporter of Yki activity, is elevated in *ex* mutant clones (**F–F'''**) but is not detectably affected in *Mer* mutant clones (**G–G'''**). Quantification is shown in (**H**). (**I–J'**) Endogenous Kib:GFP

*Figure 1 continued on next page*

*Figure 1 continued*

levels are elevated in single *Mer* somatic mosaic clones (**I and I'**) and in double *sd; Mer* clones (**J and J'**). Yellow dashed lines indicate clone boundaries. All scale bars=20 µm. Quantification in (**C**) and (**H**) is represented as the mean ± standard error of the mean (SEM); n=number of clones (no more than two clones per wing disc were used for quantification). Statistical analysis was performed using nonparametric Mann–Whitney U-test. Throughout the paper, statistical significance is reported as follows: ***p≤0.001, **p≤0.01, *p≤0.05, ns (not significant, p>0.05).

type cells in *Mer* clones (*Figure 1D–E'''*). These results indicate that Ex is more potent at inhibiting Yki nuclear translocation than Mer, consistent with Ex's ability to limit Yki activity by direct sequestration at the junctional cortex (*Badouel et al., 2009*) and suggesting that loss of *ex* should have a greater effect on pathway target gene expression than loss of *Mer*.

To compare the effects of *Mer* and *ex* loss on target gene expression, we examined the expression of *ban3>GFP* (*Matakatsu and Blair, 2012*), a reporter for one of Yki's target genes *bantam* (*Thompson and Cohen, 2006*; *Nolo et al., 2006*). *ban3>GFP* expression was significantly upregulated in *ex* mutant clones, whereas no detectible difference was observed in *Mer* clones relative to control tissue (*Figure 1F–H*), indicating that Yki is more active in *ex* clones than in *Mer* clones. Together, these results suggest that the dramatic increase in Kib levels in *Mer* clones cannot be explained strictly by Yki-mediated transcriptional feedback and that Kib is also regulated via a previously unrecognized non-transcriptional mechanism.

## Hpo pathway components regulate Kib abundance non-transcriptionally

If a Yki-independent mechanism is responsible for Kib upregulation in *Mer* clones, then Kib levels should be elevated in *Mer* clones in the absence of Yki activity. To test this hypothesis, we took advantage of a previously published method of blocking Yki-mediated transcription downstream of the Hpo pathway by removing Yki's DNA-binding partner, Sd, in the eye imaginal disc, where Sd is dispensable for cell viability (*Koontz et al., 2013*; *Yu and Pan, 2018*). Endogenous Kib::GFP was upregulated in *sd; Mer* double-mutant clones to a similar degree as in *Mer* single-mutant clones (*Figure 1I–J'*), suggesting that Mer regulates Kib levels independently of Yki activity.

To understand how Mer regulates Kib levels, we set out to develop a simpler approach to uncouple Kib protein abundance from its transcriptional regulation. Recently, the *ubiquitin 63E* promoter was used to drive expression of other Hpo pathway components to study their post-translational regulation (*Aerne et al., 2015*; *Fulford et al., 2019*), based on the assumption that the ubiquitin promoter is not regulated by Yki activity. Therefore, we made a transgenic fly line ectopically expressing Kib-GFP-FLAG under control of the ubiquitin promoter (Ubi>Kib-GFP) (*Figure 2A*). Similar to endogenous Kib::GFP, Ubi>Kib-GFP localized both at the junctional and medial cortex (*Figure 2—figure supplement 1A*). Flies expressing Ubi>Kib-GFP had slightly undergrown wings compared to control flies expressing Ubi>GFP (*Figure 2—figure supplement 1B*), suggesting that Ubi>Kib-GFP promotes Hpo pathway activity. Although wild-type flies expressing Ubi>Kib-GFP were viable, Ubi>Kib-GFP only partially rescued the *kibra^{del}* null allele (*Yu et al., 2010*), suggesting that expression from the *Ubiquitin* promoter may not be sufficient in some tissues that require Kib for viability.

Consistent with the hypothesis that Mer negatively regulates Kib levels non-transcriptionally, depletion of Mer in the posterior compartment of the wing disc using the *hh>Gal4* driver led to a substantial increase in Ubi>Kib-GFP levels across the entire compartment (*Figure 2B*). Knockdown of Sav, Hpo, and Wts also dramatically increased Ubi>Kib-GFP levels (*Figure 2C–E*), suggesting that regulation of Kib abundance is not mediated uniquely by Mer but is Hpo pathway-dependent. In contrast, expression of a Ubi>RFP control transgene was not affected by depletion of Hpo, confirming that Yki does not regulate expression at the ubiquitin promoter (*Figure 2—figure supplement 1C–C''*). Ex is also upregulated upon Hpo pathway inactivation, with a particularly strong increase when Hpo or Wts is depleted (*Hamaratoglu et al., 2006*; *Figure 2—figure supplement 2A–D*). Ex and Kib also form a complex in cultured cells (*Genevet et al., 2010*; *Yu et al., 2010*), raising the possibility that the increase in Ubi>Kib-GFP levels upon Hpo or Wts depletion is caused by increased interaction with Ex resulting in greater Kib stability. To test this possibility, we compared Ex and Ubi>Kib-GFP levels in *hpo* or *sd; hpo* double-mutant clones. While Ubi>Kib-GFP levels were similarly elevated in both *hpo* and *sd; hpo* double-mutant clones (*Figure 2F–F'''* and *Figure 2—figure*

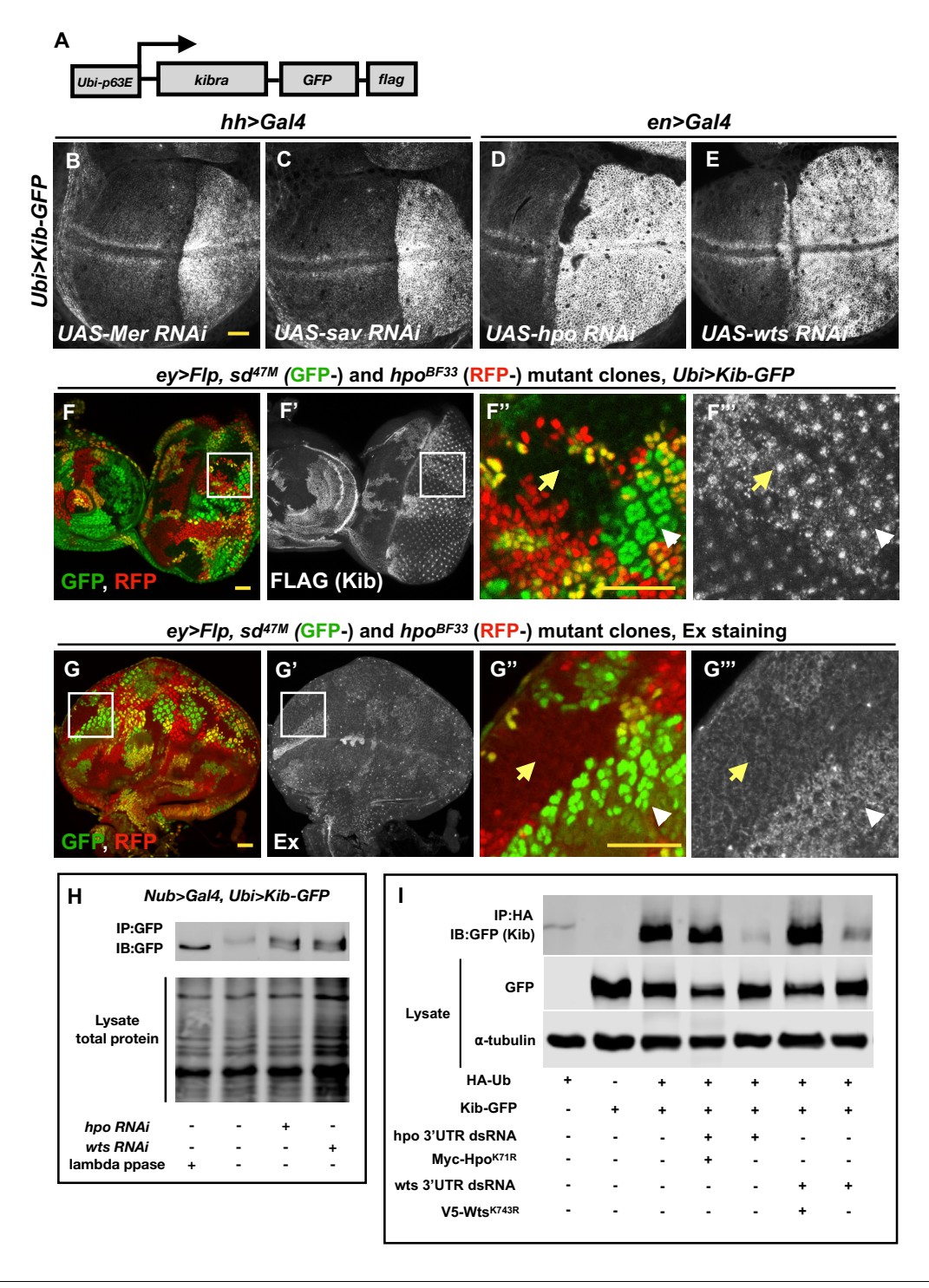

**Figure 2.** The Hippo (Hpo) pathway regulates Kibra (Kib) levels independently of Yorkie (Yki)-mediated transcription. (**A**) A cartoon of the DNA construct used to generate the *Ubi>Kib-GFP* transgenic fly line. (**B–E**) Depletion of Hpo pathway components Mer, Sav, Hpo, and Wts by RNAi in the posterior compartment of the wing results in elevated Kib-GFP levels. All scale bars=20 μm. Throughout the paper, wing imaginal discs are oriented with posterior side to the right and dorsal side up. (**F–F'''**) In the eye imaginal disc, Kib-GFP is upregulated both in *hpo* mutant clones and *sd; hpo* double-mutant clones, indicating that Hpo pathway activity controls Kib levels independently of Yki/Sd-mediated transcription. White arrowheads indicate *hpo* single-mutant clones; yellow arrows indicate *sd; hpo* double-mutant clones. Note: the clonal GFP marker (*sd+*), which is nuclear,

*Figure 2 continued on next page*

*Figure 2 continued*

is readily distinguishable from Kib-GFP, which is apical. (G–G''') Ex levels are also upregulated in *hpo* mutant clones; but in contrast to Kib, Ex upregulation is not observed in *sd; hpo* double-mutant clones. (H) Kib is phosphorylated in wing discs, and depletion of Hpo or Wts leads to decreased Kib phosphorylation. (I) Kib is ubiquitinated in S2 cells. Depletion of Hpo or Wts with dsRNA targeting 3'-untranslated region (UTR) of each kinase leads to decreased Kib ubiquitination; the effect of Hpo or Wts knockdown is rescued by addition of kinase-dead Hpo$^{K71R}$ or Wts$^{K743R}$. Throughout the paper, all immunoblot data are representative of at least three replicates.

The online version of this article includes the following figure supplement(s) for figure 2:

**Figure supplement 1.** The Hippo pathway regulates Kibra (Kib) levels independently of Yorkie (Yki) transcriptional activity.

**Figure supplement 2.** The Hippo (Hpo) pathway regulates Kibra (Kib) levels independently of Ex.

---

*supplement 2E*), Ex levels were upregulated only in *hpo* single-mutant clones but not in *sd; hpo* double-mutant clones (*Figure 2G–G'''*), indicating that the increase in Kib levels upon Hpo pathway inactivation is not mediated via Ex. Furthermore, transient co-depletion of Hpo and Yki in the wing disc posterior compartment using Gal80$^{ts}$ did not suppress the increase in Kib abundance observed when Hpo alone was depleted, even though Yki was sufficiently depleted to suppress tissue overgrowth induced by Hpo depletion alone (*Figure 2—figure supplement 2F*). Together, these results provide strong evidence that the Hpo pathway regulates Kib levels independently of Yki transcriptional output.

## The Hpo pathway promotes Kib phosphorylation and ubiquitination

Our observation that Hpo pathway activity controls Kib levels in a Yki-independent manner suggests that Kib could be regulated post-translationally. Protein abundance is commonly regulated by phosphorylation-dependent ubiquitination, and multiple Hpo pathway components are regulated via ubiquitin-mediated proteasomal degradation (*Ribeiro et al., 2014*; *Rodrigues-Campos and Thompson, 2014*; *Cao et al., 2014*; *Aerne et al., 2015*; *Ma et al., 2018*; *Ly et al., 2019*). Therefore, we hypothesized that the Hpo pathway could promote Kib phosphorylation and target it for ubiquitination and subsequent degradation.

We first asked whether Kib is phosphorylated in a pathway-dependent manner in vivo. To this end, we examined the phosphorylation state of Kib-GFP in wing imaginal discs depleted for either Hpo or Wts using a gel shift assay. In wild-type controls, phosphatase treatment of immunoprecipitated Kib-GFP resulted in increased mobility and coalescence into a single band, suggesting that Kib is normally phosphorylated (*Figure 2H*). Depletion of either Hpo or Wts resulted in a faster migrating Kib band that aligned with phosphatase-treated Kib (*Figure 2H*), suggesting that Kib is phosphorylated in a pathway-dependent manner in vivo.

Next, we asked if Kib is ubiquitinated and, if so, whether this depends on Hpo pathway activity. To address this question, we expressed Kib-GFP and hemagglutinin (HA)-tagged ubiquitin in cultured *Drosophila* Schneider 2 (S2) cells. We found that Kib was ubiquitinated and that depletion of the core pathway kinases Hpo or Wts resulted in dramatically decreased Kib ubiquitination (*Figure 2I*). Taken together, these results suggest that Kib is phosphorylated and ubiquitinated in a Hpo pathway-dependent manner and that these post-translational modifications promote its degradation.

## Slimb regulates Kib levels via a consensus degron motif

To better understand how the Hpo pathway controls Kib levels via ubiquitination, we sought to identify the machinery that mediates this process. Protein ubiquitination occurs via an enzymatic cascade that culminates in the covalent attachment of ubiquitin molecules to substrates by E3 ubiquitin ligases (*Zheng and Shabek, 2017*). We first tested the effects of depletion or overexpression of E3 ubiquitin ligases previously reported to act within the Hpo pathway on Ubi>Kib-GFP abundance. Of these, only depletion of the F-box protein Slimb, and its partners SkpA and Cul1, increased Ubi>-Kib-GFP levels (*Figure 3A and A'* and *Figure 3—figure supplement 1A–D*). Importantly, increased Ubi>Kib-GFP was evident throughout the affected cells in comparison to control tissue (*Figure 3A'*), suggesting that overall Kib abundance was increased. Because loss of Slimb increases Ex levels

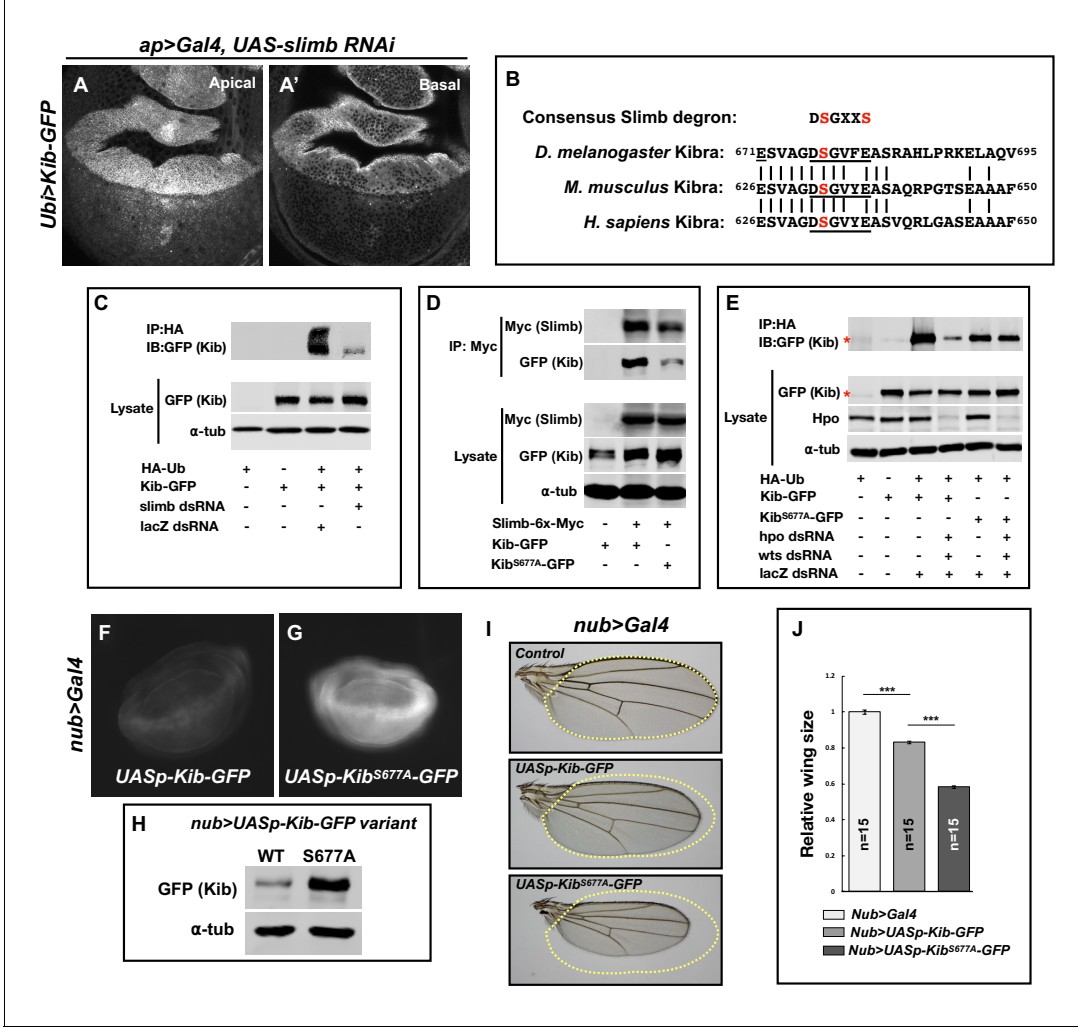

**Figure 3.** Slimb regulates Kibra (Kib) abundance via a consensus degron. (A–A') Depletion of Slimb in the dorsal compartment of the wing imaginal disc results in increased Kib-GFP levels both apically (A) and basally (A'). (B) Alignment of the fly, mouse, and human Kib protein sequences showing the conservation of the putative Slimb degron motif DSGXXS (underlined). The vertical lines indicate conserved residues. (C) Immunoblot showing that depletion of Slimb in S2 cells decreases Kib ubiquitination. (D) Co-IP experiments showing that Kib forms a complex with Slimb in S2 cell lysates in a degron-dependent manner. (E) Ubiquitination of the degron mutant, Kib$^{S677A}$, is diminished and is insensitive to Hippo pathway inactivation. Asterisks indicate non-specific bands. (F–G) Widefield fluorescence images of wing discs expressing either *UASp-Kib-GFP* (F) or *UASp-Kib$^{S677A}$-GFP* (G) with the *nub>Gal4* driver; images were taken using identical settings. (H) Immunoblot of wing disc cell lysates (20 discs each) of *UASp-Kib-GFP* or *UASp-Kib$^{S677A}$-GFP* expressed with the *nub>Gal4* driver. (I–J) Ectopic expression of Kib$^{S677A}$-GFP in the wing results in stronger growth suppression than expression of wild-type Kib-GFP. Quantification of wing sizes in (I) is represented as mean ± SEM relative to the control; n=number of wings (one wing per fly). Statistical comparison was performed using the one-way analysis of variance (ANOVA) test followed by Tukey's Honestly Significant Difference (HSD) test.

The online version of this article includes the following figure supplement(s) for figure 3:

**Figure supplement 1.** Effect of different E3 ubiquitin ligases involved in the Hippo (Hpo) pathway on Kibra (Kib) levels.

(*Ribeiro et al., 2014*) and Ex interacts with Kib in cultured cells (*Genevet et al., 2010*; *Yu et al., 2010*), we considered the possibility that increased Ubi>Kib-GFP upon Slimb depletion could result indirectly from ectopic interactions with increased Ex. However, co-depletion of Ex and Slimb did not suppress the increase in Ubi>Kib-GFP levels (*Figure 3—figure supplement 1E–G'*), suggesting that Slimb directly regulates Kib abundance.

Slimb is a homolog of the mammalian β-TrCP that functions as a substrate-targeting component of the SCF E3 ubiquitin ligase complex by recognizing a consensus degron motif on target proteins (*Skaar et al., 2013*). Kib contains a conserved single stretch of amino acids $^{676}$DSGVFE$^{681}$ that

matches the consensus Slimb degron (*Figure 3B*). If Slimb regulates Kib stability via the degron, then we predict that (1) Kib ubiquitination should be Slimb-dependent, (2) Slimb should physically interact with Kib via the degron, (3) mutation of the degron site should diminish Kib ubiquitination, and (4) the degron mutant Kib should display greater stability than wild-type Kib. Using S2 cells, we found that depletion of Slimb severely reduces Kib ubiquitination and that Kib and Slimb formed a complex (*Figure 3C and D*). Additionally, mutating a serine residue in Kib (Kib$^{S677A}$) known to be important for proper substrate recognition by Slimb (*Hart et al., 1999*; *Rogers et al., 2009*; *Morais-de-Sá et al., 2013*; *Ribeiro et al., 2014*) significantly reduced both Slimb–Kib interaction (*Figure 3D*) and Kib ubiquitination (*Figure 3E*).

To assess the effects of the degron mutation on protein stability in vivo, we generated wild-type and Kib$^{S677A}$ transgenes inserted at identical genomic positions and expressed under control of the upstream activating sequence (UAS). For these experiments, we used the UASp promoter (*Rørth, 1998*), which expresses at lower levels in somatic tissues than UASt (attempts to generate a transgenic line expressing Kib$^{S677A}$ under the ubiquitin promoter were unsuccessful, presumably because ubiquitous expression of a stabilized form of Kib is lethal). Kib$^{S677A}$-GFP accumulated to much greater levels than wild-type Kib-GFP when expressed in the wing disc pouch using the *nub>-Gal4* driver (*Figure 3F–H*). Confocal imaging revealed that while Kib-GFP and Kib$^{S677A}$-GFP had similar localizations apically, Kib$^{S677A}$-GFP displayed bright foci in basal tissue sections (*Figure 3—figure supplement 1H–I'*), presumably due to protein aggregation caused by higher Kib levels. Consistent with the observed increased protein abundance, expression of Kib$^{S677A}$-GFP under the *nub>-Gal4* driver led to significantly smaller adult wings than did wild-type Kib-GFP (*Figure 3I and J*). We presume this was because of increased Kib-driven upstream pathway activity, though we have not demonstrated this directly. Collectively, these results indicate that Slimb regulates Kib stability in vivo.

## The Hpo pathway regulates Kib abundance via Slimb

To this point, our results identify both the Hpo pathway and Slimb as regulators of Kib abundance, but they do not resolve whether the two mechanisms act in parallel or together. We reasoned that if Slimb regulates Kib levels in parallel to the Hpo pathway, then loss of pathway components in tissue expressing Kib$^{S677A}$ would have an additive effect on Kib levels. Conversely, if Hpo pathway components regulate Kib abundance via Slimb, then Kib$^{S677A}$ should be insensitive to pathway inactivation. We first tested the effect of depleting Hpo pathway components on ubiquitination of Kib$^{S677A}$. In striking contrast to wild-type Kib, ubiquitination of Kib$^{S677A}$ was not sensitive to depletion of Hpo and Wts (*Figure 3E*), suggesting that the Hpo pathway promotes Kib degradation via Slimb-mediated ubiquitination.

To test if the Hpo pathway promotes Kib degradation via Slimb in vivo, we induced *Mer* mutant clones in wing imaginal discs expressing either wild-type Kib-GFP or Kib$^{S677A}$-GFP under the *nub>-Gal4* driver. Similar to endogenous Kib (*Figure 1B*) or Kib expressed by the ubiquitin promoter (*Figure 2B*), UASp-Kib-GFP was dramatically upregulated apically and basally in *Mer* clones relative to control cells (*Figure 4A–C' and G*). In contrast, Kib$^{S677A}$-GFP appeared only mildly apically stabilized in *Mer* clones (*Figure 4D–E'*), with no detectable difference in basal Kib$^{S677A}$-GFP levels between the clone and control cells (*Figure 4F–F' and H*). Taken together, these results indicate that the Hpo pathway regulates Kib levels via the degron motif. Interestingly, in *Mer* clones but not in control cells, Kib-GFP also formed bright aggregate-like foci basally (*Figure 4C*), similar to Kib$^{S677A}$-GFP (*Figure 3—figure supplement 1I'*), suggesting these foci form as a result of high Kib levels.

The slight apical stabilization of Kib$^{S677A}$-GFP in *Mer* clones could be caused by two possibilities that are not mutually exclusive: (1) Slimb could still weakly bind Kib$^{S677A}$-GFP and promote its degradation, albeit with reduced efficiency, and (2) loss of Hpo pathway activity could lead to greater cortical Kib accumulation at the expense of the total cytoplasmic pool. In support of the first possibility, Kib$^{S677A}$-GFP weakly associated with Slimb (*Figure 3D*) and was still slightly ubiquitinated in S2 cells (*Figure 3E*). To ask whether the mild apical accumulation of Kib$^{S677A}$-GFP in *Mer* clones could also be caused by cortical recruitment, we examined Kib in tissues lacking Hpo, which resulted in stronger junctional accumulation of Ubi>Kib-GFP than loss of Mer (*Figure 2D*). Strikingly, whereas wild-type Kib-GFP increased both apically and basally in *hpo* clones (*Figure 4I–K' and O*), Kib$^{S677A}$-GFP increased apically but decreased basally in *hpo* clones (*Figure 4L–N' and P*). These results suggest

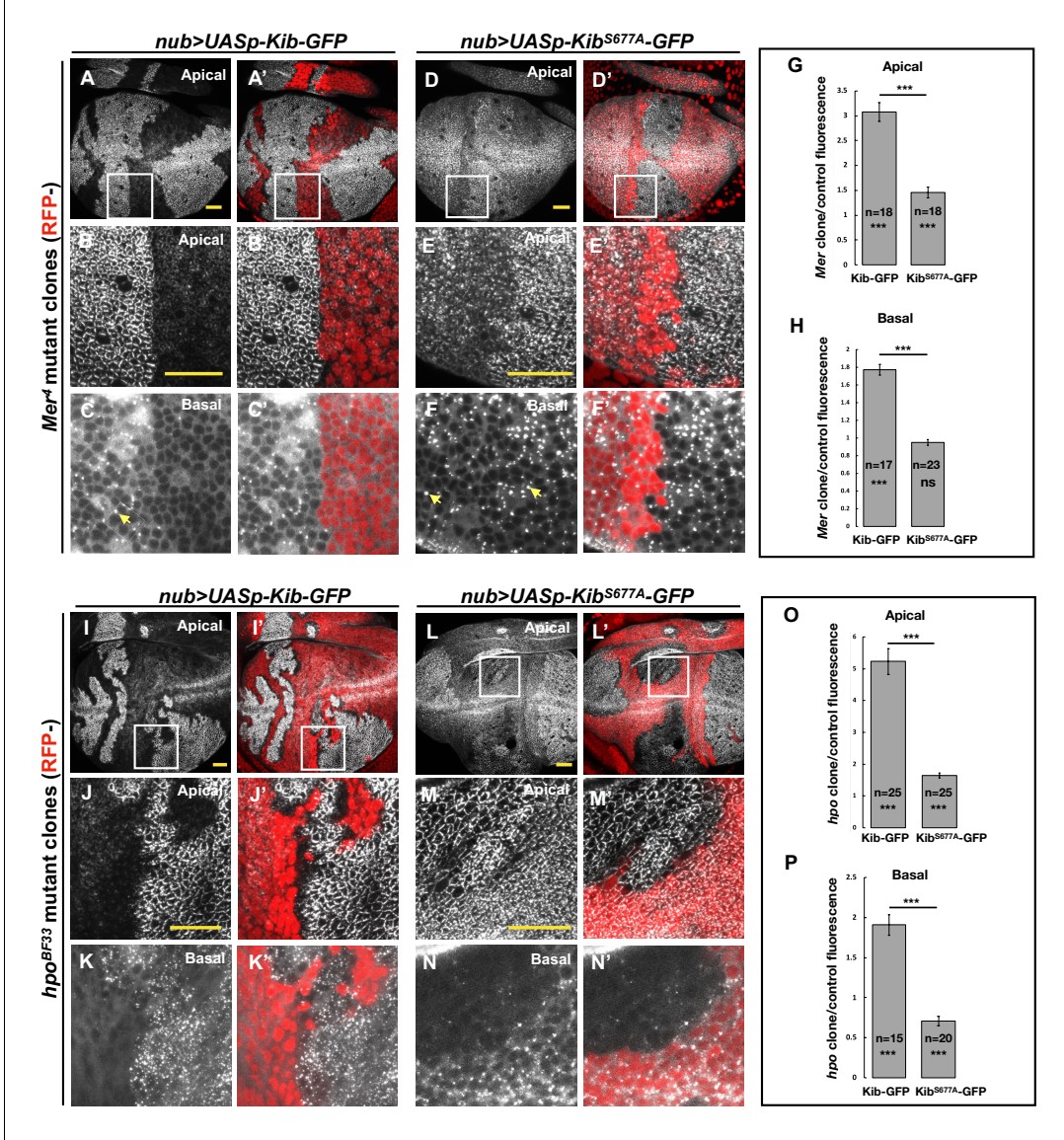

**Figure 4.** The Hippo pathway regulates Kibra (Kib) abundance via a putative degron motif. (**A–F'**) *Mer* somatic mosaic clones in wing discs expressing either *UASp-Kib-GFP* (**A-C'**) or *UASp-Kib*$^{S677A}$*-GFP* (**D–F'**) with the *nub>Gal4* driver. Note that wild-type Kib-GFP is significantly elevated in *Mer* clones both apically and basally, while Kib$^{S677A}$-GFP is only slightly stabilized apically and is not affected basally. Yellow arrows in C and F point to presumed Kib aggregates due to increased abundance. All scale bars=20 µm. (**G–H**) Quantification of clone/control ratio of apical (**G**) and basal (**H**) Kib-GFP fluorescence. All quantification is represented as the mean ± SEM; asterisks above the plots show p-values between the transgenes; asterisks inside each bar show p-values for each transgene with respect to 1; n=number of clones (no more than two clones per wing disc were used for quantification). Statistical comparison was performed using Mann–Whitney *U*-test. (**I–N'**) *hpo* somatic mosaic clones in wing discs expressing either *UASp-Kib-GFP* (**I–K'**) or *UASp-Kib*$^{S677A}$*-GFP* (**L–N'**) with the *nub>Gal4* driver. Note that wild-type Kib-GFP levels are significantly elevated in *hpo* clones both apically and basally, while Kib$^{S677A}$-GFP is stabilized apically but depleted basally in *hpo* clones. (**O–P**) Quantification of clone/control ratio of apical (**O**) and basal (**P**) Kib-GFP fluorescence.

that the stabilization of Kib$^{S677A}$-GFP observed upon Hpo pathway inactivation is, at least in part, due to the recruitment of Kib apically, where it might be stabilized in a protein complex.

## The Hpo pathway promotes Kib degradation in a highly compartmentalized manner and independently of pathway activation by Ex

Previous work showed that Ex interacts with Kib in S2 cells and suggested that Kib and Ex function in a complex to regulate the Hpo pathway (*Yu et al., 2010*; *Genevet et al., 2010*). In contrast, in vivo studies suggest that Kib functions in parallel to Ex and its partner Crb to regulate activity of the downstream kinase cascade (*Baumgartner et al., 2010*; *Yu et al., 2010*; *Su et al., 2017*). Given these observations, we wondered whether loss of Ex or Crb would result in elevated Ubi>Kib-GFP abundance similar to the loss of Mer, Sav, Hpo, or Wts. To our surprise, depletion of Ex and Crb, either individually or together, had no detectable effect on Ubi>Kib-GFP levels (*Figure 5A–C'*). Moreover, reducing Hpo pathway activity by other means, such as by overexpressing Dachs or depleting Fat, Ds, or the Hpo activator Tao-1 (*Boggiano et al., 2011*), similarly had no effect on Ubi>Kib-GFP levels (*Figure 5—figure supplement 1A–E*). On the other hand, knockdown of Mats' or Kib's binding partner, Pez (*Poernbacher et al., 2012*), increased Ubi>Kib-GFP levels (*Figure 5—figure supplement 1F–G*). These results suggest that upstream regulation of the Hpo pathway is highly compartmentalized and that Kib degradation is promoted specifically via the pathway components it associates with during Hpo pathway activation.

This parallel behavior of Hpo pathway regulation prompted us to ask whether increasing the activity of one upstream branch of the pathway can substitute for the loss of another. To test this idea, we asked if the Ubi>Kib-GFP transgene, which causes mild undergrowth in a wild-type background (*Figure 2—figure supplement 1E*), can suppress the lethality of $ex^{e1}$, a null allele (*Boedigheimer and Laughon, 1993*). Ubi>Kib-GFP strongly suppressed $ex^{e1}$ lethality, producing viable and fertile adult flies at expected frequencies (*Figure 5—figure supplement 1I*) that completely lacked Ex protein (*Figure 5—figure supplement 1J–K'*). Homozygous $ex^{e1}$; *Ubi>-Kib-GFP/+* flies had significantly larger wings than heterozygotes (*Figure 5D–E*), but otherwise were phenotypically

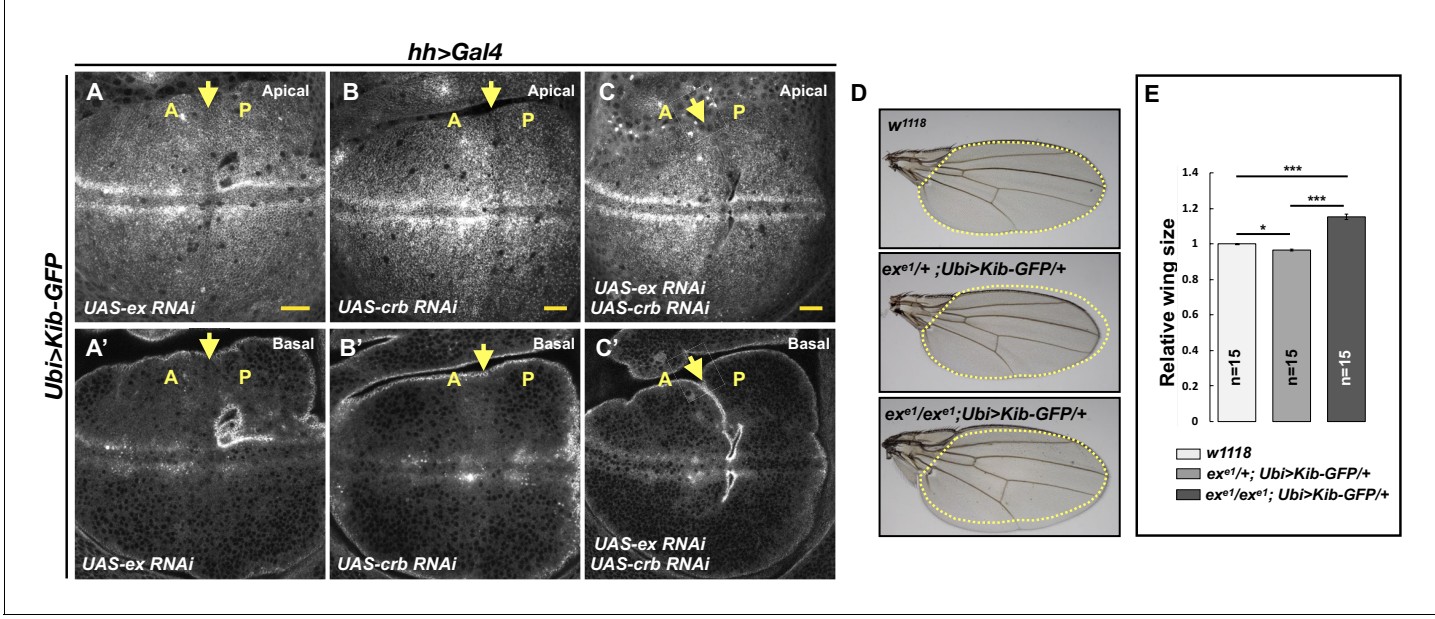

**Figure 5.** Kibra (Kib) abundance is regulated independently of Ex. (A–C') Depletion of Ex (A and A'), Crumbs (Crb; B and B'), or both Ex and Crb (C and C') in the posterior wing imaginal disc does not affect Ubi>Kib-GFP abundance. Yellow arrows indicate the anterior–posterior (A–P) boundary. Scale bars=20 μm. (D–E) Adult wings of $w^{1118}$, $ex^{e1}/+$; *Ubi>Kib-GFP/+*, or $ex^{e1}/ex^{e1}$; *Ubi>Kib-GFP/+* flies. Quantification of wing sizes in (E) is represented as the mean ± SEM; n=number of wings (one wing per fly). Statistical comparison was performed using the one-way ANOVA test followed by Tukey's HSD test.

The online version of this article includes the following figure supplement(s) for figure 5:

**Figure supplement 1.** The Hippo (Hpo) pathway controls Kibra (Kib) abundance in a tightly compartmentalized manner.

normal. Together, these results establish that Kib and Ex signal in parallel to regulate at least some aspects of pathway activity.

## The WW domains of Kib are essential for its degradation via the Hpo pathway and Slimb

Our discovery that Kib degradation is tightly compartmentalized suggests that complex formation between Kib and other Hpo pathway components might be an important step both for pathway activation and Kib degradation. Indeed, Kib interacts with Sav, Mer, Hpo (via Sav), and Wts in S2 cells (*Baumgartner et al., 2010*; *Genevet et al., 2010*; *Yu et al., 2010*) and can recruit these components to the apical cell cortex in vivo (*Su et al., 2017*). To test this idea, we first asked if the pathway kinases Hpo and Wts play a structural vs. an enzymatic role in promoting Kib ubiquitination. For these experiments, kinase-dead versions of Hpo or Wts (Hpo$^{K71R}$ and Wts$^{K743R}$, respectively; *Wu et al., 2003*; *Huang et al., 2005*) were transfected into cells depleted of endogenous Hpo or Wts with dsRNA targeted against their 3'-UTRs (the kinase-dead constructs lacked endogenous UTRs). To our surprise, expression of Hpo$^{K71R}$ or Wts$^{K743R}$ restored Kib ubiquitination when the endogenous kinases were depleted, indicating that these kinases promote Kib ubiquitination via complex formation rather than phosphorylation (*Figure 2I*).

Next, we performed a structure/function analysis to map the region in Kib that could mediate complex formation and promote its degradation by the Hpo pathway components. Kib is a multivalent adaptor protein that contains at least seven potential functional regions: two N-terminal WW domains (WW1 and WW2), a C2-like domain, a putative atypical protein kinase C (aPKC)-binding domain, and three coiled-coil regions (CC1, CC2, and CC3; *Figure 6A*). We generated transgenic fly lines expressing different truncations of Kib-GFP under control of the ubiquitin promoter. Two transgenes, one expressing Kib lacking the C2-like domain and another encoding the first 483 amino acids (aa) of Kib, produced sterile transformants and could not be maintained as stable lines. The rest of the transgenes produced viable and fertile flies.

A Kib truncation lacking the C-terminal third of the coding sequence (Kib$^{1-857}$-GFP) but retaining the degron motif was strongly upregulated upon Hpo depletion, similar to wild-type Kib (*Figure 6—figure supplement 1A–B'*). Flies expressing Kib$^{1-857}$-GFP had smaller wings than those expressing wild-type Kib-GFP (*Figure 6—figure supplement 1K*), suggesting that deletion of the C-terminal region enhances Kib activity. In contrast, a Kib truncation lacking the first 483 aa (Kib$^{484-1288}$-GFP) was insensitive to Hpo depletion even though it retained the Slimb degron motif (*Figure 6—figure supplement 1D and D'*), suggesting that the degron alone is not sufficient for pathway-mediated degradation of Kib. Interestingly, Kib$^{484-1288}$-GFP was much less potent at suppressing wing growth compared to wild-type Kib-GFP (*Figure 6—figure supplement 1K*), indicating that the first 483 amino acids of Kib are also essential for Hpo pathway activation.

The first 483 amino acids of Kib contain two WW domains, as well as CC1 and CC2 regions (*Figure 6A*). Deletion of either CC1 or CC2 did not prevent Kib upregulation upon Hpo depletion, indicating that these regions do not mediate pathway-dependent Kib degradation (*Figure 6—figure supplement 1E–F'*). However, Kib variants lacking the WW domains, either individually (Kib$^{\Delta WW1}$-GFP and Kib$^{\Delta WW2}$-GFP) or together (Kib$^{\Delta WW1\&2}$-GFP), expressed at markedly higher levels than wild-type Kib (*Figure 6B*). Additionally, these proteins accumulated at the junctional cortex and appeared to be depleted basally but were not upregulated when Hpo was depleted (*Figure 6C–C''* and *Figure 6—figure supplement 1G–I'*). Thus, the WW domains of Kib are necessary for its degradation via the Hpo pathway. Importantly, while Kib lacking the WW domains interacted with Slimb normally in S2 cells (*Figure 6—figure supplement 2A*), depletion of Slimb had no effect on Kib$^{\Delta WW1}$-GFP levels (*Figure 6D–D''*), again suggesting that association between Kib and Slimb alone is not sufficient for Kib degradation.

Further characterization of the WW domain truncations revealed differences in effects on growth and subcellular localization (*Figure 6—figure supplement 1K–O*). Kib$^{\Delta WW1\&2}$-GFP often had an extremely punctate appearance in imaginal tissues (*Figure 6—figure supplement 1N–O'*). Adult flies expressing Kib$^{\Delta WW1\&2}$-GFP were homozygous viable and had wings almost the size of $w^{1118}$ controls (*Figure 6—figure supplement 1K*) despite the fact that it expressed at higher levels (*Figure 6B*). Kib$^{\Delta WW2}$-GFP also had a punctate appearance (*Figure 6—figure supplement 1M and M'*), but adults expressing Kib$^{\Delta WW2}$-GFP had significantly smaller wings than flies expressing wild-type Kib-GFP (*Figure 6—figure supplement 1K*). Deletion of WW1 (Kib$^{\Delta WW1}$-GFP) resulted in a

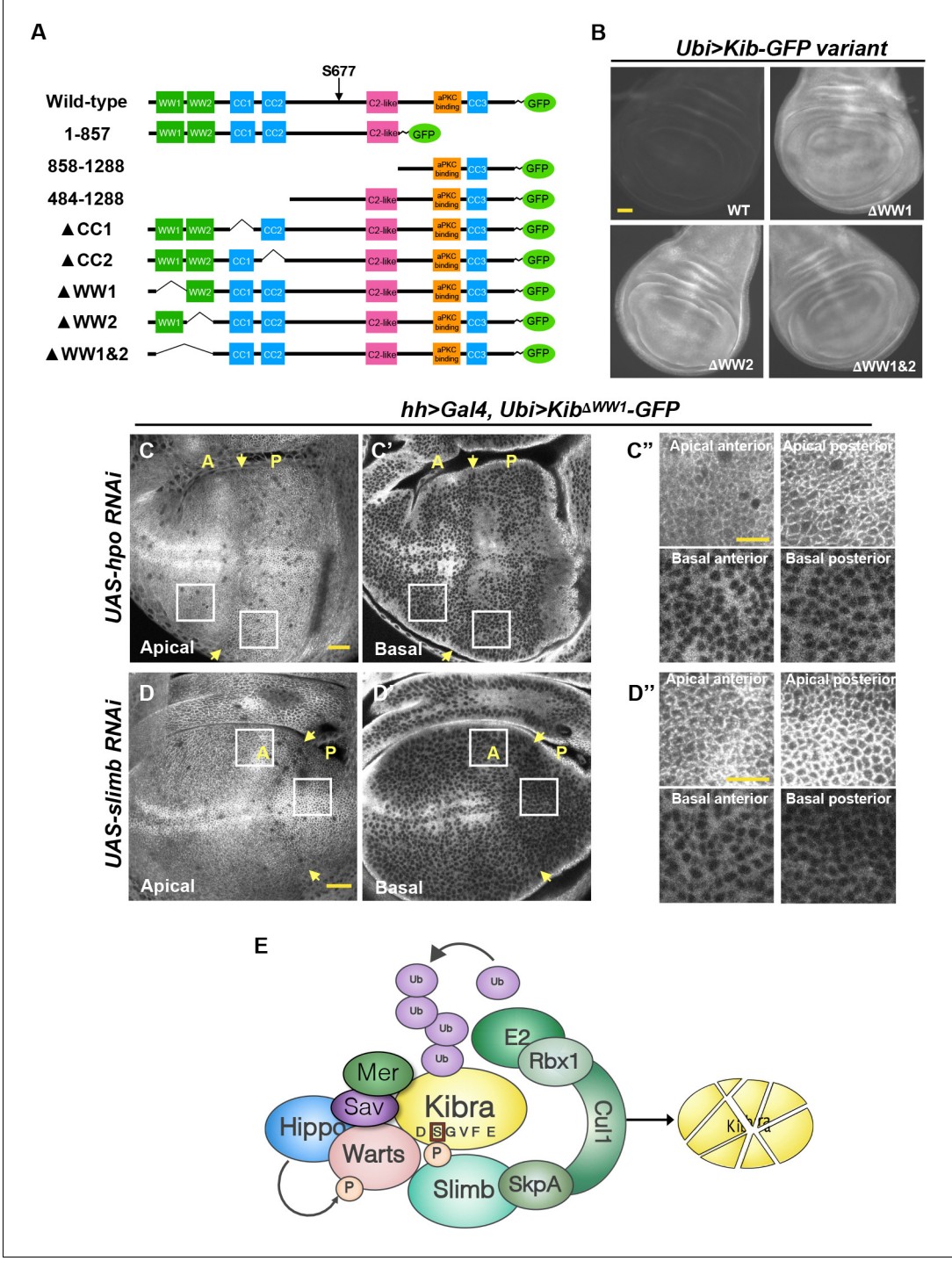

**Figure 6.** The WW domains of Kibra (Kib) are required for Hippo (Hpo) pathway- and Slimb-mediated degradation. (**A**) Diagram of Kib truncations generated for this study. (**B**) Widefield fluorescence images of wing imaginal discs expressing wild-type and WW-domain truncations of Kib-GFP expressed under the ubiquitin promoter. All images were taken with identical settings. Scale bar=40 μm. (**C–C''**) Depletion of Hpo does not affect expression of *Ubi>Kib^{ΔWW1}-GFP*. Note that Hpo depletion leads to apical stabilization and basal depletion of Kib^{ΔWW1}-GFP (**C''**). (**D–D''**) Depletion of Slimb does not affect expression of *Ubi>Kib^{ΔWW1}-GFP*. Note that similar to Hpo depletion, loss of Slimb leads to slight apical stabilization and basal depletion of Kib^{ΔWW1}-GFP (**D''**). Yellow arrows indicate A–P boundary of the wing discs. Scale bars=20 μm (**C and D**) and 10 μm (**C'' and D''**). (**E**) A model of Kib degradation by the Hpo pathway and Slimb.

The online version of this article includes the following figure supplement(s) for figure 6:

*Figure 6 continued on next page*

*Figure 6 continued*
**Figure supplement 1.** The role of WW domains in Hippo pathway-mediated Kibra (Kib) degradation.
**Figure supplement 2.** Complex formation and Kibra (Kib) degradation.

protein that localized at the apical cortex but did not form puncta (*Figure 6—figure supplement 1L and L'*). Flies expressing Kib$^{\Delta WW1}$-GFP had wings equal in size to $w^{1118}$ control (*Figure 6—figure supplement 1K*). Taken together, these results indicate that while both WW domains are required for pathway-mediated Kib turnover, only the WW1 domain of Kib is necessary for Hpo pathway activation.

We reasoned that if complex formation between Kib and other Hpo pathway components is necessary for Kib degradation, then Slimb might also be a part of this complex. Consistent with this prediction, Slimb co-immunoprecipitated with Mer, Hpo, and Wts in S2 cells (*Figure 6—figure supplement 2B–D*). We then asked whether the role of the WW domains in Kib degradation was to mediate Kib interaction with other Hpo pathway components. A previous study found that deletion of both WW domains enhanced Kib interaction with Mer in S2 cells (*Baumgartner et al., 2010*), a result we confirmed (*Figure 6—figure supplement 2E*). Kib interacts with Wts in flies (*Genevet et al., 2010*; *Yu et al., 2010*), and mammalian Kib interacts with Lats2 (a mammalian homolog of Wts) via the WW domains (*Xiao et al., 2011*). We found that the interaction of Kib$^{\Delta WW1\&2}$ with Wts was significantly weakened (*Figure 6—figure supplement 2F*). Additionally, it was previously reported that Pez interacts with Kib via the WW domains in S2 cells (*Poernbacher et al., 2012*), consistent with our in vivo observation that loss of Pez leads to higher Kib levels. Collectively, these results suggest that pathway-mediated Kib degradation requires the WW domains of Kib, possibly because these domains mediate Kib interaction with multiple pathway components.

## Mechanical tension patterns Kib degradation across the wing disc epithelium

We next sought to address the potential developmental significance of Kib degradation by the Hpo pathway and Slimb. Observation of wing discs ectopically expressing either *UASp-Kib-GFP* or *UASp-Kib$^{S677A}$-GFP* revealed strikingly different patterns of Kib abundance throughout the tissue. Ectopically expressed wild-type Kib-GFP appeared more abundant at the center of the wing pouch with a marked decrease in fluorescence at the tissue periphery (*Figure 7A and A'*). A similar pattern of abundance was observed for endogenously expressed Kib::GFP (*Figure 7—figure supplement 1A*). In contrast, ectopically expressed Kib$^{S677A}$-GFP fluorescence was distributed more uniformly throughout the wing pouch (*Figure 7B and B'* and *Figure 7—figure supplement 1B–D*). Because both transgenes were expressed from identical genomic locations and under the same ectopic promoter, we reasoned that the difference between Kib and Kib$^{S677A}$ abundance throughout the tissue was likely a result of differential protein turnover.

If the abundance of Kib$^{S677A}$-GFP is disproportionately higher at the periphery of the wing blade, which corresponds to the proximal regions of the adult wing, then that region should display more severe growth defects when compared to wild-type Kib-GFP. To ask whether growth was disproportionately inhibited in the proximal region of the wing, we first measured the wing aspect ratios comparing the width of the proximal or distal wing regions to the overall proximal-distal length. Strikingly, while the relative decrease in width distally was mild in *nub>UASp-Kib-GFP* or *nub>UASp-Kib$^{S677A}$-GFP* wings compared to control wings, the proximal width decreased dramatically in wings expressing Kib$^{S677A}$, indicating that expression of Kib$^{S677A}$ inhibited growth disproportionately more in the proximal region (*Figure 7C*). Similarly, when the wing length was measured in the proximal-distal (P–D) axis, using L4 vein as an estimate of total length and the posterior crossvein as the approximated midpoint, we found that wing growth was more severely inhibited proximally than distally (*Figure 7D*). Collectively, these results suggest that Kib degradation occurs in a patterned manner in the wing imaginal epithelium and could serve to pattern growth of this tissue.

The pattern of Kib degradation we observe, higher at the periphery and lower in the center, is similar to the pattern of junctional tension in the wing blade reported previously (*Legoff et al., 2013*; *Mao et al., 2013*). This similarity raises the possibility that mechanical tension patterns Kib

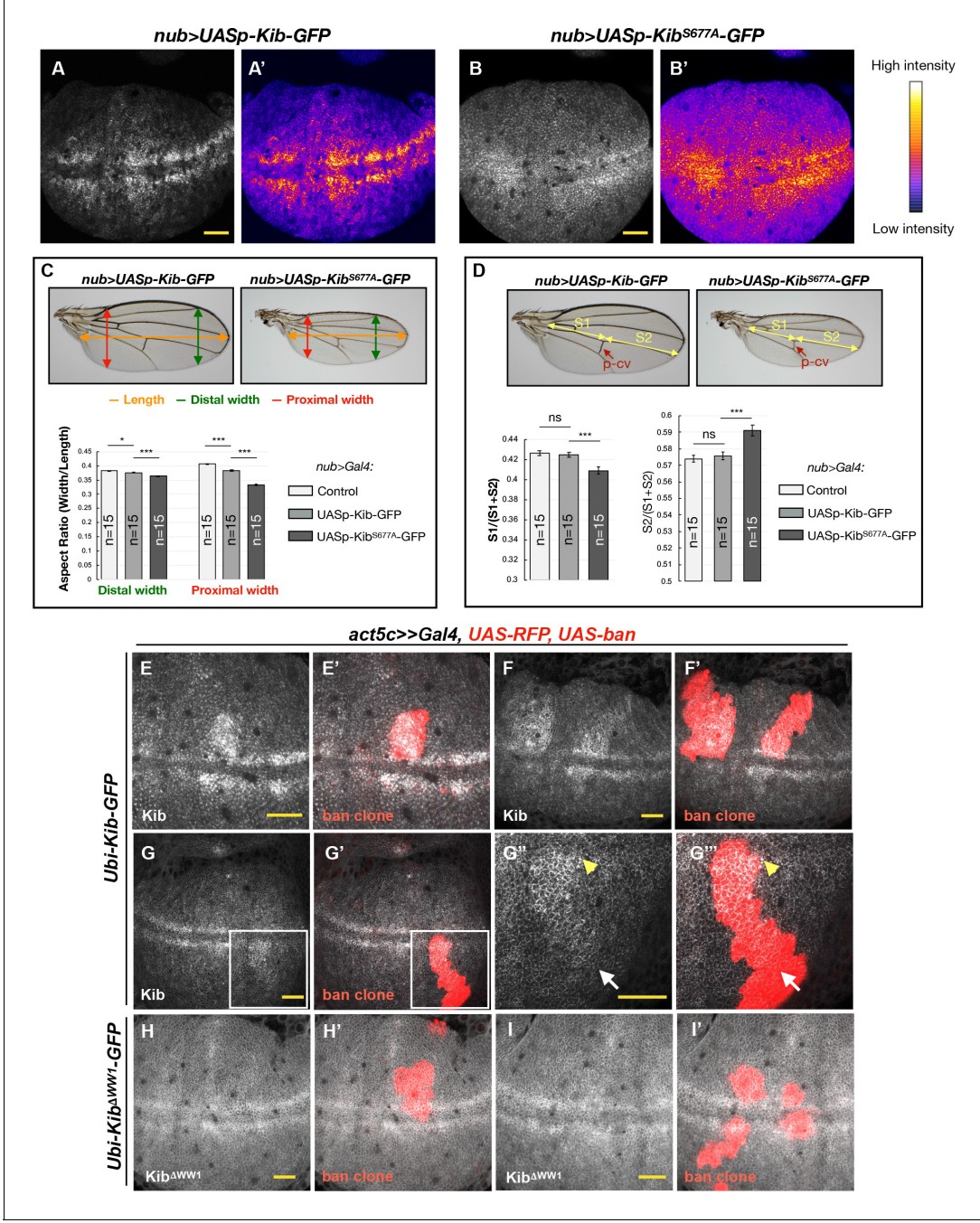

**Figure 7.** Kibra (Kib) degradation is patterned by mechanical tension in the wing pouch to control proportional growth. (A–B′) Grayscale images of the wing pouch, which produces the adult wing blade, expressing *UASp-Kib-GFP* (A) or *UASp-Kib^S677A^-GFP* (B) at identical genomic locations under the *nub>Gal4* driver. Corresponding heatmap intensity images are shown in (A′) and (B′). Note that Kib^S677A^-GFP displays a more uniform distribution across the pouch than wild-type Kib-GFP. (C) Quantification of aspect ratios of adult wings expressing *nub>Gal4* alone or with *UASp-Kib-GFP* and *UASp-Kib^S677A^-GFP*. The color-coded segments in the wing image represent the wing length (orange), distal width (green), and proximal width (red). (D) Quantification of the length of proximal (S1) or distal (S2) wing region with respect to total wing length in wings expressing *nub>Gal4* alone or with *UASp-Kib-GFP* and *UASp-Kib^S677A^-GFP*; p-cv=posterior crossvein. All quantification is represented as the mean ± SEM; n=number of wings (one wing per fly). Statistical comparison was performed using the one-way ANOVA test followed by Tukey's HSD test. (E–F′) Kib-GFP levels are elevated in rapidly proliferating *UAS-bantam* clones. (G–G′′′) Increased Kib abundance is more pronounced at the center of the wing pouch (yellow arrowhead) than at its periphery (white arrow). (H–I′) Kib^ΔWW1^-GFP levels do not change in *bantam*-expressing clones. All scale bars=20 μm.

The online version of this article includes the following figure supplement(s) for figure 7:

**Figure supplement 1.** Kibra (Kib) degradation is not uniform across the pouch region of the wing imaginal disc.

degradation to regulate pathway activity in parallel to previously described tension-sensing mechanisms that regulate pathway output (*Rauskolb et al., 2014*; *Deng et al., 2015*; *Alégot et al., 2019*). As an initial test of this hypothesis, we used a previously described method of reducing tension in the wing imaginal epithelium that uses somatic mosaic clones expressing the growth promoting miRNA gene *bantam* (*Pan et al., 2016*). These clones proliferate faster than and therefore are compressed by the surrounding wild-type cells, leading to lower junctional tension within the clones and higher Hpo pathway activity (*Pan et al., 2016*). Indeed, we observed higher levels of Kib-GFP within *bantam* clones (*Figure 7E–G'''*). Interestingly, the increase in Kib abundance in *bantam*-expressing clones was stronger near the center of the wing pouch than its periphery (*Figure 7G–G'''*), consistent with previous observations that the compression within *bantam*-expressing clones is greater at the center of the pouch than at the periphery, presumably because cells near the center of the wing pouch are already more compressed (*Pan et al., 2016*). Importantly, Kib$^{\Delta WW1}$-GFP levels did not change in *bantam* clones (*Figure 7H–I'*), as expected if tension-induced Kib degradation requires Kib-mediated Hpo signaling complex formation. Taken together, these results suggest that Kib degradation is patterned by mechanical tension across the wing pouch resulting in decreased Kib in regions of high tension and greater Yki promoted growth.

## Discussion

In this study, we show that the Hpo pathway negatively regulates Kib levels via a previously unrecognized post-translational feedback loop. Several key results indicate that this feedback is independent of Yki transcriptional activity. First, loss of Mer leads to a dramatic increase in Kib levels without a detectable increase in Yki transcriptional activity. Second, removing Sd, which blocks Yki-mediated transcription, does not suppress the elevated Kib levels upon Hpo pathway inactivation. Third, the abundance of Kib-GFP expressed under a Yki-insensitive promoter (*Ubi>Kib-GFP*) still increases upon Hpo pathway inactivation. Additionally, we show that Kib is phosphorylated and ubiquitinated in a pathway-dependent manner, and that Kib ubiquitination is mediated via SCF$^{Slimb}$.

Multiple upstream components regulate the core Hpo kinase cassette, but their organization and the degree of crosstalk between them has not been well elucidated. A striking aspect of our findings is the extent to which Kib degradation by the Hpo pathway is insulated from the activity of other upstream pathway regulators. Previous studies have shown that Crb and Ex function together at the junctional cortex (*Chen et al., 2010*; *Ling et al., 2010*; *Robinson et al., 2010*; *Sun et al., 2015*) and in parallel to Kib at the medial cortex (*Su et al., 2017*). Ft can influence Ex stability at the junctions (*Silva et al., 2006*; *Wang et al., 2019*), suggesting linkage between these Hpo signaling branches, and Tao-1 functions downstream of Ex (*Chung et al., 2016*). Although Ex can form a complex with Kib in S2 cells (*Genevet et al., 2010*; *Yu et al., 2010*) and Kib junctional localization is dependent on Crb (*Su et al., 2017*), our results show that depletion of Crb, Ex, Ft, or Tao-1 does not affect Kib-GFP levels. Together with our results that Hpo and Wts promote Kib ubiquitination independently of their kinase activity (*Figure 2I*), these data strongly suggest that Kib degradation is triggered by Kib-mediated complex formation. Given the importance of the putative phospho-degron in Kib turnover, we propose that Kib is phosphorylated at S677 upon formation of the signaling complex by an as yet unidentified kinase, leading to the recruitment of SCF$^{Slimb}$ and Kib ubiquitination. This model potentially explains the striking compartmentalization of Kib degradation that we observe—loss of Ex or Crb has no effect on Kib abundance because they do not participate in Kib-mediated signaling complexes. Compartmentalized, parallel regulation of pathway activity could clearly have functional implications for the control of tissue growth but at the moment is poorly understood.

The precise dynamics that lead to Hpo pathway activation vs. Kib degradation remain to be uncovered. In the simplest scenario, we propose that the same complex can function to repress Yki activity or target Kib for degradation. If so, the impact of Kib-mediated complex formation on overall pathway activity could be altered by the relative dynamics of Slimb-mediated degradation vs. Yki phosphorylation. As a consequence, this mechanism could provide a means for regulation of pathway output by factors outside of the pathway itself.

A remaining question our work defines relates to the functional significance of this mechanism to regulate Kib abundance in developing tissues. Our results suggest that Kib-mediated Hpo signaling is patterned across the wing imaginal epithelium by regulated Kib degradation. Specifically, Kib degradation via the degron-dependent mechanism we have identified is greater at the periphery of the

wing pouch than at its center. Previous studies have shown that junctional tension also is greater at the periphery of the wing pouch (*Legoff et al., 2013*; *Mao et al., 2013*). Higher tension at the wing pouch periphery has been proposed to drive tissue growth, possibly by promoting Yki function, as a compensatory mechanism for low levels of diffusible morphogens that are expressed in narrow bands of cells at the center of the pouch (*Shraiman, 2005*; *Aegerter-Wilmsen et al., 2007*; *Hariharan, 2015*; *Pan et al., 2018*). Interestingly, junctional tension is known to repress Hpo pathway activity via inhibition of Wts (Lats1/2 in mammals; *Rauskolb et al., 2014*; *Ibar et al., 2018*), though it remains unknown whether junctional tension also affects upstream Hpo pathway regulators such as Kib. Given our results, we propose that tension also promotes Kib degradation and thereby reduces Kib-mediated upstream pathway activation.

It might seem paradoxical that Kib degradation, which is dependent on signaling complex assembly, is greater at the periphery of the wing pouch where net Hpo pathway activity is thought to be lower (*Hariharan, 2015*; *Pan et al., 2018*). However, our results using kinase-dead forms of Hpo and Wts clearly suggest that Kib ubiquitination and degradation can be uncoupled from activation of pathway kinases. We imagine that tension might regulate access of the degradation components (e.g. the putative kinase or SCF$^{Slimb}$) to Kib-organized signaling complexes and thereby regulate Kib-mediated pathway activation. We currently know little about how dynamic Hpo pathway output is in developing tissues, largely because there are no available single-cell resolution reporters for pathway activity. Our findings suggest that pathway output mediated by Kib could be tightly and dynamically regulated in response to mechanical tension or other factors that affect the degradation mechanism described here.

Another question our study raises is the functional significance of having both transcriptional and post-translational negative feedback mechanisms that regulate Kib levels. Feedback regulation is a common feature in cell signaling, and transcriptional negative feedback can serve to limit the output of a signaling pathway over time (*Perrimon and McMahon, 1999*). In the case of the Hpo pathway, transcriptional feedback mediated by Yki is not specific to Kib, as Ex and Mer expression is also promoted by Yki activity. Moreover, loss of any upstream Hpo pathway regulator, including Ex, Crb, Ft, or Tao-1 would presumably affect Kib levels via the transcriptional feedback. In contrast, the post-translational feedback identified in this study would silence Kib function in a more rapid and specific manner. The role of the post-translational feedback could be to enhance the robustness of Kib-mediated signaling (*Stelling et al., 2004*), possibly by preventing drastic fluctuations in Kib levels, to ensure optimally scaled and patterned tissue growth. On a broader level, our identification of Kib-specific feedback highlights the importance of understanding why there are multiple upstream inputs regulating the Hpo pathway and how they function during development.

# Materials and methods

## Key resources table

| Reagent type (species) or resource | Designation | Source or reference | Identifiers | Additional information |
|---|---|---|---|---|
| Gene (*Drosophila melanogaster*) | *kibra* | DOI: 10.1016/j.devcel.2009.12.012<br>DOI: 10.1016/j.devcel.2009.12.011<br>DOI: 10.1016/j.devcel.2009.12.013 | FLYB: FBgn0262127 | |
| Genetic reagent (*D. melanogaster*) | Kib::GFP | DOI: 10.1016/j.devcel.2017.02.004 | | |
| Genetic reagent (*D. melanogaster*) | *Mer⁴ 19AFRT* | *LaJeunesse et al., 1998* | | |
| Genetic reagent (*D. melanogaster*) | *exᵉ¹ 40AFRT* | PMID: 8269855 | | |
| Genetic reagent (*D. melanogaster*) | *19AFRT sd⁴⁷ᴹ* | *Wu et al., 2008* | | |

*Continued on next page*

*Continued*

| Reagent type (species) or resource | Designation | Source or reference | Identifiers | Additional information |
|---|---|---|---|---|
| Genetic reagent (*D. melanogaster*) | *hpo^{BF33} 42DFRT* | *Jia et al., 2003* | | |
| Genetic reagent (*D. melanogaster*) | *ban3-GFP* | DOI: 10.1242/dev.070367 | | |
| Genetic reagent (*D. melanogaster*) | *UAS-Mer RNAi* | DOI: 10.1016/j.devcel.2017.02.004 | | |
| Genetic reagent (*D. melanogaster*) | *UAS-sav RNAi* | Bloomington *Drosophila* Stock Center | BL 28006 | |
| Genetic reagent (*D. melanogaster*) | *UAS-hpo RNAi* | Vienna *Drosophila* Resource Center | VDRC 104169 | |
| Genetic reagent (*D. melanogaster*) | *UAS-wts RNAi* | Vienna *Drosophila* Resource Center | VDRC 106174 | |
| Genetic reagent (*D. melanogaster*) | *UAS-ex RNAi* | Vienna *Drosophila* Resource Center | VDRC 109281 | |
| Genetic reagent (*D. melanogaster*) | *UAS-crb RNAi* | Vienna *Drosophila* Resource Center | VDRC 39177 | |
| Genetic reagent (*D. melanogaster*) | *UAS-yki RNAi (III)* | Vienna *Drosophila* Resource Center | VDRC 40497 | |
| Genetic reagent (*D. melanogaster*) | *UAS-slimb RNAi* | Bloomington *Drosophila* Stock Center | BL 33898 | |
| Genetic reagent (*D. melanogaster*) | *UAS-Cul1 RNAi* | Bloomington *Drosophila* Stock Center | BL 29520 | |
| Genetic reagent (*D. melanogaster*) | *UAS-SkpA RNAi* | Bloomington *Drosophila* Stock Center | BL 32870 | |
| Genetic reagent (*D. melanogaster*) | *UAS-mahj RNAi* | Bloomington *Drosophila* Stock Center | BL 34912 | |
| Genetic reagent (*D. melanogaster*) | *UAS-Nedd4 RNAi* | Bloomington *Drosophila* Stock Center | BL 34741 | |
| Genetic reagent (*D. melanogaster*) | *UAS-POSH RNAi* | Bloomington *Drosophila* Stock Center | BL 64569 | |
| Genetic reagent (*D. melanogaster*) | *UAS-POSH* | Bloomington *Drosophila* Stock Center | BL 58990 | |
| Genetic reagent (*D. melanogaster*) | *UAS-Su(dx) RNAi* | Bloomington *Drosophila* Stock Center | BL 67012 | |
| Genetic reagent (*D. melanogaster*) | *UAS-Herc4* | DOI: 10.1371/journal.pone.0131113 | | |

*Continued on next page*

*Continued*

| Reagent type (species) or resource | Designation | Source or reference | Identifiers | Additional information |
|---|---|---|---|---|
| Genetic reagent (*D. melanogaster*) | *UAS-Smurf RNAi* | Bloomington *Drosophila* Stock Center | BL 40905 | |
| Genetic reagent (*D. melanogaster*) | *UAS-Fbxl7 RNAi* | Vienna *Drosophila* Resource Center | VDRC 108628 | |
| Genetic reagent (*D. melanogaster*) | *UAS-ft RNAi* | Bloomington *Drosophila* Stock Center | BL 34970 | |
| Genetic reagent (*D. melanogaster*) | *UAS-ds RNAi* | Vienna *Drosophila* Resource Center | VDRC 36219 | |
| Genetic reagent (*D. melanogaster*) | *UAS-dachs-V5* | DOI: 10.1242/dev.02427 | | |
| Genetic reagent (*D. melanogaster*) | *UAS-Tao1 RNAi* | Vienna *Drosophila* Resource Center | VDRC 17432 | Previously used in DOI: 10.1016/j.devcel.2011.08.028 |
| Genetic reagent (*D. melanogaster*) | *UAS-mats RNAi* | Bloomington *Drosophila* Stock Center | BL 34959 | |
| Genetic reagent (*D. melanogaster*) | *UAS-Pez RNAi* | Bloomington *Drosophila* Stock Center | BL 33918 | |
| Genetic reagent (*D. melanogaster*) | *Ey>Flp 19AFRT Ubi-GFP; Ubi-RFP 42DFRT* | DOI: 10.1016/j.devcel.2013.04.021 | | |
| Genetic reagent (*D. melanogaster*) | *Ft-GFP* | VDRC 318477 | | |
| Genetic reagent (*D. melanogaster*) | *Ds:GFP* | **Brittle et al., 2012** | | |
| Genetic reagent (*D. melanogaster*) | *Ubi-Kib-GFP-FLAG 86Fb* | This paper | | See Materials and methods section |
| Genetic reagent (*D. melanogaster*) | *UASp-Kib-GFP-FLAG 86Fb* | This paper | | See Materials and methods section |
| Genetic reagent (*D. melanogaster*) | *UASp-Kib$^{S677A}$-GFP-FLAG 86Fb (this study)* | This paper | | See Materials and methods section |
| Genetic reagent (*D. melanogaster*) | *Ubi-Kib-GFP-FLAG VK37* | This paper | | See Materials and methods section |
| Genetic reagent (*D. melanogaster*) | *Ubi-KibΔWW1-GFP-FLAG VK37* | This paper | | See Materials and methods section |
| Genetic reagent (*D. melanogaster*) | *Ubi-KibΔWW2-GFP-FLAG VK37* | This paper | | See Materials and methods section |
| Genetic reagent (*D. melanogaster*) | *Ubi-KibΔWW1 and 2-GFP-FLAG VK37* | This paper | | See Materials and methods section |

*Continued on next page*

*Continued*

| Reagent type (species) or resource | Designation | Source or reference | Identifiers | Additional information |
|---|---|---|---|---|
| Genetic reagent (*D. melanogaster*) | *Ubi-Kib1-857-GFP-FLAG VK37* | This paper | | See Materials and methods section |
| Genetic reagent (*D. melanogaster*) | *Ubi-Kib484-1288-GFP-FLAG VK37* | This paper | | See Materials and methods section |
| Genetic reagent (*D. melanogaster*) | *Ubi-Kib858-1288-GFP-FLAG VK37* | This paper | | See Materials and methods section |
| Genetic reagent (*D. melanogaster*) | *Ubi-KibΔCC1-GFP-FLAG VK37* | This paper | | See Materials and methods section |
| Genetic reagent (*D. melanogaster*) | *Ubi-KibΔCC2-GFP-FLAG VK37* | This paper | | See Materials and methods section |
| Antibody | anti-Ex (Guinea pig polyclonal) | DOI: 10.1016/j.cub.2006.02.063 | RRID:AB_2568722 | Tissue staining (1:5000) |
| Antibody | anti-FLAG (Mouse monoclonal) | Sigma Aldrich | Cat#F1804; RRID:AB_262044 | IB (1:20,000) |
| Antibody | anti-Sd (Guinea pig polyclonal) | *Guss et al., 2013* | RRID:AB2567874 | Tissue staining (1:1000) |
| Antibody | anti-GFP (Guinea pig polyclonal) | DOI: 10.1091/mbc.E19-07-0387 | NA | IP (1:1250) |
| Antibody | anti-GFP (Rabbit polyclonal) | Michael Glotzer (University of Chicago) | NA | IB (1:5000) |
| Antibody | anti-Hpo (mouse polyclonal) | DOI: 10.1016/j.devcel.2017.02.004 | NA | IB (1:5000) |
| Antibody | anti-HA (Rabbit polyclonal) | Santa Cruz | Cat#sc-805; RRID:AB_631618 | IB (1:5000) |
| Antibody | anti-Myc 9B11 (Mouse monoclonal) | Cell Signaling | Product #2276 | IP (1:1000) IB (1:40,000) |
| Antibody | anti-V5 (Mouse monoclonal) | GenScript | Cat# A01724-100 | IB (1:2500) |
| Antibody | anti-alpha tubulin (Mouse monoclonal) | Sigma Aldrich | Cat# T 9026 | IB (1:2500) |
| Cell line (*D. melanogaster*) | S2-DGRC | Cherbas Lab, Indiana University | RRID:CVCL_TZ72 | https://dgrc.bio.indiana.edu/product/View?product=6 |

## Fly genetics

For expression of UAS transgenes, the following drivers were used: *hh>Gal4, en>Gal4, ap>Gal4, nub>Gal4*.

To generate mutant clones, the following crosses were performed:

## Kib::GFP in *ex* or *Mer* mutant clones

y w hsFlp; Ubi-RFP 40A FRT X ex$^{e1}$ 40A FRT/CyO, dfdYFP; Kib::GFP/TM6, Tb

Mer$^4$ 19A FRT/FM7, actGFP; MKRS/TM3, Ser, actGFP X hsFLP, w$^{1118}$, Ubi-RFP-nls 19AFRT; Kib::GFP/TM3, Ser, actGFP

## Yki-YFP in *ex* or *Mer* mutant clones

> y w hsFlp; Ubi-RFP 40A FRT X ex$^{e1}$ Yki-YFP yki$^{B5}$/CyO, dfdYFP
> Mer$^4$ 19A FRT/+; Yki-YFP/CyO, dfdYFP X hsFLP, w$^{1118}$, Ubi-RFP-nls 19AFRT;MKRS/TM3, Ser, actGFP

## *ban3-GFP* in *ex* or *Mer* mutant clones

> y w hsFlp; Ubi-RFP 40A FRT X ex$^{e1}$ 40A FRT/CyO, dfdYFP; ban3-GFP/TM6, Tb
> Mer$^4$ 19A FRT/FM7, actGFP; MKRS/TM3, Ser, actGFP X hsFLP, w$^{1118}$, Ubi-RFP-nls 19AFRT; ban3-GFP/TM3, Ser, actGFP

## Kib::GFP in *sd Mer* double-mutant clones

> sd$^{47}$ Mer$^4$ 19A FRT/FM7, dfdYFP; Sco/CyO, dfdYFP X hsFLP, w$^{1118}$, Ubi-RFP-nls 19AFRT; Kib::GFP/TM3, Ser, actGFP

## Ubi>Kib-GFP in single *sd* or *hpo* mutant clones or in *sd hpo* double-mutant clones

> sd$^{47}$ 19A FRT/FM7, dfdYFP; FRT 42D hpo$^{BF33}$/CyO, dfdYFP X ey>Flp Ubi-GFP 19A FRT; FRT 42D Ubi-RFP/CyO, dfdYFP; Ubi>Kib-GFP/+

## UASp-Kib-GFP or UASp-Kib$^{S677A}$-GFP in *Mer* or *hpo* mutant clones

> Mer$^4$ 19A FRT/+; nub>Gal4/CyO, dfdYFP X hsFLP, w$^{1118}$, Ubi-RFP-nls 19AFRT; UASp-Kib-GFP/TM3, Ser, actGFP
> Mer$^4$ 19A FRT/+; nub>Gal4/CyO, dfdYFP X hsFLP, w$^{1118}$, Ubi-RFP-nls 19AFRT; UASp-Kib$^{S677A}$-GFP/TM3, Ser, actGFP
> nub>Gal4 FRT 42D hpo$^{BF33}$/CyO, dfdYFP X y w hsFLP; FRT 42D Ubi-RFP/CyO, dfdYFP; UASp-Kib-GFP/+
> nub>Gal4 FRT 42D hpo$^{BF33}$/CyO, dfdYFP X y w hsFLP; FRT 42D Ubi-RFP/CyO, dfdYFP; UASp-Kib$^{S677A}$ -GFP/+

## Expression constructs and generation of *Drosophila* transgenic lines

To generate Ubi>Kib-GFP, Kib was fused to GFP-FLAG with a linker sequence 5′-TCCGGTACCGGCTCCGGC-3′, and the entire Kib-GFP-FLAG cassette was first cloned into UAStattB backbone to generate UASt-Kib-GFP-FLAG, with unique NotI (immediately 5′ of the Kozak sequence) and KpnI (in the linker region) restriction sites flanking Kib sequence. To make Kib$^{1-857}$, Kib$^{858-1288}$, and Kib$^{484-1288}$, the corresponding regions were amplified (*Supplementary file 1*); UAStattB was linearized with NotI and KpnI and the amplified fragments were cloned into linearized backbone via Gibson assembly (*Gibson et al., 2009*). Fragments lacking CC or WW domains were made using an inverse PCR approach with flanking primers (*Supplementary file 1*) and the amplified linear pieces including the plasmid backbone were circularized via Gibson assembly. Kib-GFP-FLAG cassettes (full-length or truncations) were amplified using flanking primers (*Supplementary file 1*) and cloned via Gibson assembly into *p63E-ubiquitin* backbone (*Munjal et al., 2015*) linearized with NotI and XbaI. The transgenes were inserted at the 86Fb (full-length Kib) or VK37 (full-length and truncated Kib) docking site via phiC31-mediated site-specific integration.

pMT-Kib-GFP-FLAG was generated by cloning Kib-GFP-FLAG cassette via Gibson assembly (Gibson to pMT primers, *Supplementary file 1*) into the pMT backbone (*Klueg et al., 2002*) linearized by KpnI and EcoRV.

UASp-Kib$^{S677A}$-GFP-FLAG was generated using Q5 Site-Directed Mutagenesis Kit (New England Biolabs, catalog #E0554S) using primers KibS677A (*Supplementary file 1*). pMT-Kib-GFP-FLAG was used as a template due to smaller size of the plasmid. The mutant Kib$^{S677A}$-GFP-FLAG cassette was excised with NotI and XbaI and ligated into pUASp (*Rørth, 1998*) to generate UASp-Kib$^{S677A}$-GFP.

Both UASp-Kib$^{S677A}$-GFP and UASp-Kib-GFP were inserted at the 86Fb docking site via phiC31-mediated site-specific integration.

## Immunostaining of imaginal tissues

In *Figure 2F-G'''*, *Figure 2—figure supplement 2A–E*, and *Figure 5—figure supplement 1J-K'*, wing or eye imaginal discs from wandering late third instar larvae were fixed and stained as previously described (*McCartney and Fehon, 1996*). Primary antibodies, listed in Key Resources table, were diluted as follows: anti-Ex (1:5000), anti-FLAG (1:20,000), and anti-Sd (1:1000). Secondary antibodies (diluted 1:1000) were from Jackson ImmunoResearch Laboratories. Immunostaining samples were imaged using either a Zeiss LSM 800 or LSM 880 confocal microscope and the images were analyzed with *Image J*.

## Live imaging of imaginal tissues

Throughout the paper (except in *Figure 2F–G'''* and *Figure 2—figure supplement 2A–E*), live tissues were used for imaging. Live imaging of the *Drosophila* imaginal tissues was performed as previously described (*Xu et al., 2019*). Briefly, freshly dissected wing or eye imaginal discs from third instar larvae were pipetted into a ~40 µl droplet of Schneider's *Drosophila* Medium supplemented with 10% fetal bovine serum and mounted on a glass slide. To support the tissue, spherical glass beads (Cospheric, Product ID: SLGMS-2.5) of ~50 µm in diameter were placed under the cover slip. The mounted samples were immediately imaged on Zeiss LSM 880 or LSM 800 confocal microscopes. Throughout the paper, apical tissue views were shown as maximum projections of the most apical optical sections (0.75 µm/section, four to five sections) generated using *Image J*; for basal views, single sections ~10.5 µm below the apical surface were shown. Widefield fluorescence imaging of live wing imaginal discs was done using a Zeiss Axioplan 2ie microscope with an Orca ER camera and Zeiss AxioVision software.

## Co-immunoprecipitation from S2 cells

The following constructs were used in co-immunoprecipitation experiments: *pMT-Kib-GFP-FLAG* (this study), *pMT-Kib$^{\Delta WW1\&2}$-GFP-FLAG* (this study), *pAc5.1-Slimb-6x-myc* (from J. Chiu, UC Davis), *pAFW-Mer*, *pAHW-Mer$^{1-600}$*, *pMT-FLAG-Hpo*, and *pAC5.1-V5-Wts* (*Huang et al., 2005*).

Briefly, $3.5 \times 10^6$ S2 cells (S2-DGRC) were transfected with total of 500 ng of the indicated DNA constructs using dimethyldioctadecylammonium bromide (Sigma; *Han, 1996*) at 250 µg/ml in six-well plates. Immunoprecipitation (IP) was performed 3 days after transfection. For expression of pMT constructs, 700 µM $CuSO_4$ was added to the wells 24 hr prior to cell lysis (2 days after transfection). For GFP or Myc IPs, guinea pig anti-GFP (1:1250) or mouse anti-Myc (1:1000) antibodies were used. Pierce Protein A (Thermo Scientific) magnetic beads were used to precipitate antibody-bound target proteins. For immunoblotting, the following antibody concentrations were used: rabbit anti-GFP (1:5000), mouse anti-Hpo (1:5000), mouse anti-α-tubulin (1:2500), mouse anti-Myc (1:40,000), mouse M2 anti-Flag (1:20,000), mouse anti-V5 (1:2500), and rabbit anti-hemagglutinin (HA) (1:5000). Immunoblots were scanned using an Odyssey CLx scanner (LI-COR Biosciences).

Cells were harvested and lysed on ice in buffer containing 25 mM Hepes, 150 mM NaCl, 1 mM ethylenediaminetetraacetic acid, 0.5 mM ethylene glycol-bis(β-aminoethyl ether)-N,N,N',N'-tetraacetic acid, 0.9 M glycerol, 0.1% Triton X-100, 0.5 mM Dithiothreitol, and Complete protease inhibitor cocktail (Roche) at one tablet/10 ml concentration.

For detection of phosphorylated Kib in vivo, dissected wing discs from wandering third-instar larvae (200 discs per condition) expressing *nub>Gal4* with Ubi>Kib-GFP alone or together with an indicated RNAi transgene were immediately flash-frozen in a bath of dry ice and 95% ethanol and stored at −80˚C. On the day of IP, the discs were briefly thawed on ice and lysed in buffer described above. PhosSTOP (Sigma Aldrich) phosphatase inhibitor cocktail was added to the lysis buffer to inhibit phosphorylation (one tablet/10 ml of buffer). Kib-GFP was immunoprecipitated with guinea pig anti-GFP antibody (1:1250). A control sample was treated with λ-phosphatase. Samples were run on 8% polyacrylamide gel, with 118:1 acrylamide/bisacrylamide (*Scheid et al., 1999*), to better resolve phosphorylated Kib species.

## Ubiquitination assay and generation of dsRNA

For ubiquitination assays, pMT-HA-Ub (*Zhang et al., 2006*) was co-transfected where indicated to provide labeled ubiquitin. To inhibit proteasomal degradation, 50 µM MG132 (Cayman Chemical) and 50 µM calpain inhibitor I (Sigma Aldrich) was added 4 hr prior to cell lysis. Cells were lysed in RIPA buffer (150 mM NaCl, 1% NP-40, 0.5% Na deoxycholate, 0.1% SDS, and 25 mM Tris [50 mM, pH 7.4]), supplemented with 5 mM N-ethylmaleimide and Complete protease inhibitor cocktail (Roche, one tablet/10 ml of buffer). HA-tagged ubiquitin was purified using Pierce anti-HA magnetic beads (clone 2–2.2.14).

For dsRNA-mediated knockdown experiments, T7 primers (*Supplementary file 1*), annealing at the 3′-UTR (for Hpo and Wts) or the coding region (for Slimb), were used to first generate polymerase chain reaction (PCR) products. The PCR products were then used as templates to transcribe dsRNA using the MEGAscript T7 Transcription Kit (ThermoFisher, catalog #13345).

## Quantification and statistical analysis

Image J was used to quantify mean fluorescence intensity in clones vs. control region in *Figure 1C, H* and *Figure 4G, H, O, and P*. In all cases, no more than two clones per imaginal disc were used for quantification. To quantify adult wing sizes, wings were mounted in methyl salicylate and photographed with the same settings on a Zeiss Axioplan 2ie microscope using a Canon camera (EOS rebel T2i). Subsequent measurements of wing size were taken using Image J. Graphical and statistical analyses were performed using MS Excel and R, respectively.

# Acknowledgements

We thank D Pan, K Irvine, J Jiang, N Tapon, D Strutt, J Chiu, M Glotzer, T Lecuit, K Guss, the Developmental Studies Hybridoma Bank, TRiP at Harvard Medical School (NIH/NIGMS R01-GM084947), the Bloomington stock center, and VDRC stock center for fly stocks and other reagents. We thank S Buiter for technical help. We thank M Glotzer and S Horne-Badovinac for helpful comments on the manuscript. S A T was supported by an NIH training grant (T32 GM007183) and an NSF-GRFP. This work was supported by a grant from the National Institutes of Health to RGF (NS034783).

# Additional information

### Funding

| Funder | Grant reference number | Author |
| --- | --- | --- |
| National Institute of Neurological Disorders and Stroke | NS034783 | Richard G Fehon |
| National Institute of General Medical Sciences | T32 GM007183 | Sherzod A Tokamov |
| National Science Foundation | Graduate Research Fellowship | Sherzod A Tokamov |

The funders had no role in study design, data collection and interpretation, or the decision to submit the work for publication.

### Author contributions

Sherzod A Tokamov, Conceptualization, Formal analysis, Supervision, Funding acquisition, Validation, Investigation, Visualization, Methodology, Writing - original draft, Writing - review and editing; Ting Su, Conceptualization, Resources, Investigation; Anne Ullyot, Validation, Investigation, Visualization, Methodology; Richard G Fehon, Conceptualization, Resources, Supervision, Funding acquisition, Validation, Project administration, Writing - review and editing

## Author ORCIDs

Sherzod A Tokamov ⬤ https://orcid.org/0000-0001-7989-1552
Ting Su ⬤ https://orcid.org/0000-0003-1124-517X
Richard G Fehon ⬤ https://orcid.org/0000-0003-4889-2602

## Decision letter and Author response

Decision letter https://doi.org/10.7554/eLife.62326.sa1
Author response https://doi.org/10.7554/eLife.62326.sa2

## Additional files

### Supplementary files

• Supplementary file 1. Primers used in this study with the corresponding sequences. Note the highlighted G residue in KibS677A For primer corresponds to the substitution that will result in the mutation of serine-677 to alanine.

• Transparent reporting form

### Data availability

All data generated or analysed during this study are included in the manuscript and supporting files.

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
