## [Decision Letter]

**Acceptance summary:**

The Hippo pathway is a highly conserved pathway controlling tissue growth during development. One of its upstream regulators is Kibra, which activates the pathway by assembling the Hippo signaling complex to regulate the activity of the downstream transcriptional effector, Yorkie. However, as shown here, activating the Hippo signaling complex also results in the phosphorylation of Kibra itself, which promotes its own degradation via the E3 ubiquitin ligase.

**Decision letter after peer review:**

Thank you for submitting your article "Yorkie-independent negative feedback couples Hippo pathway activation with Kibra degradation" for consideration by *eLife*. Your article has been reviewed by three peer reviewers, one of whom is a member of our Board of Reviewing Editors, and the evaluation has been overseen by K VijayRaghavan as the Senior Editor. The reviewers have opted to remain anonymous.

The reviewers have discussed the reviews with one another and the Reviewing Editor has drafted this decision to help you prepare a revised submission.

This is an interesting, well-written manuscript, which addresses the post-transcriptional regulation of Kibra, a regulator of the Hippo pathway, an important pathway controlling growth. The authors convincingly show that Kibra stability is regulated by Slimb, and that Hippo pathway kinases Hpo and Wts promote Kibra phosphorylation and thus Slimb-dependent degradation. They have strong biochemical and imaging-based analyses. They define a function for a Slimb degron, and conduct structure-function to explore Kibra domains in protein turnover. Overall this is an interesting, well-conducted story presenting novel data that will be of interest to the cell signaling and growth control fields. Publication in *eLife* is supported, provided the authors can clarify the points raised by the reviewers.

Summary:

Many of the points can be easily addressed by the authors, but they should particularly pay attention to their comments to Figure 7.

Reviewer #1:

In this manuscript, the authors present data to further unravel the complex regulation of the Hippo pathway, an important pathway controlling growth. Their data point to a negative feedback loop mediated by the core Hippo kinase cassette, which regulates one of its upstream regulators, Kibra, in a Yorkie-independent way. In addition, they provide some evidence that Kibra degradation is patterned across the disc. These are novel data and provide additional information on how such an important pathway is controlled to prevent excess growth of tissues, at least in some regions of the fly wing disc.

1) They show that knock-down of Sav, Hippo or Warts results in increased Ubi>Kib-GFP levels, but I do not understand why they also "… regulate its localization at the junctional cortex.". In both control and sav-, hpo- and wts-RNAi Ubi>Kib-GFP can be found at the junction (Supplementary Figure 1F).

2) Figure 2I. The authors show dramatic decrease of ubiquitinated Ubi>Kib-GFP upon hpo- and wts-RNAi, as well as less phosphorylated Ubi>Kib-GFP upon single knock-down. No explanation is given why in one experiment they used single knock down (H), but double knock-down in (I). In addition, the statement that Kib levels are regulated "via phosphorylation-dependent ubiquitination" is an assumption, although it may be correct given findings published previously.

3) Subsection “Slimb regulates Kibra levels via a consensus degron motif”: If loss of Slmb leads to increased Ex (Ribeiro et al) and loss of either Slmb or Ex leads to increased Ubi>Kib-GFP (Figure 3A and Figure 1A, respectively), why should co-depletion of Ex and Slimb "… suppress the increase in Ubi>Kib-GFP levels”?

4) Given their conclusion suggesting that the Kibra pathway is patterned, I wonder whether the authors always have studied the same region of the disc when comparing different genotypes. This is not always obvious from the figures.

Reviewer #2:

This interesting, well-written manuscript addresses the post-transcriptional regulation of the Hippo pathway regulator Kibra. The authors convincingly show that Kibra stability is regulated by Slimb, and that Hippo pathway kinases Hpo and Wts promote Kibra phosphorylation and thus Slimb-dependent degradation. They have strong biochemical and imaging based analyses. They define a function for a Slimb degron, and conduct structure function to explore Kibra domains in protein turnover. Overall this is an interesting, well conducted and novel story that is appropriate for publication in *eLife*.

Endogenous Kib ( tagged with GFP) abundance is increased in merlin clones, but not in expanded clones, even though ex clones dramatically increase nuclear yorkie and ban3>gfp, and merlin does not. These data indicate loss of ex has a more potent effect on Yki than loss of merlin, and suggest Merlin acts independent of Yki and transcription. Consistent with this model, loss of Sd does not affect Kib levels in the eye. Importantly, ubiquitin driven KibraGFP is increased by loss of Merlin, Sav, Hpo or Wts. The authors note strong junctional accumulation of Kibra upon loss of Hpo or Wts. The authors show Kibra is phosphorylated and ubiquitinated upon Hpo/Wts activation. They show that Kibra has a Slimb degron (DSGVFE), and that mutation of this motif leads to stabilization of Kibra, and makes its levels insensitive to Hpo pathway activation. They also conducted structure function analyses that indicate that the WW domains of Kibra are important for its degradation upon Hpo pathway activation. Because Slimb promotes Ex degradation, loss of Slimb leads to increased Ex, and hence increased Hpo/Wts activity. The authors show overexpressed Ex does not in itself stabilize Kibra. While it is clear that Kibra phosphorylation is promoted by Hpo/Wts, it is still unclear if this is a direct phosphorylation The authors propose a model in which Kibra promotes Hpo/Wts activity, which then locally promotes Kibra degradation. It is not clear how it this local signaling is isolated from other junctional Hpo and Wts.

The weakest part of this very strong manuscript is Figure 7. If the intent is to show that Kibra is in a pattern in the wing, it would be good to include the staining and intensity quantification of endogenous Kibra in A and B comparison. In Figure 7C and D, the authors should include pictures of examples of adult wings of the different genotypes that they provide the size analysis. Details of the landmarks on the wings they used to define distal and proximal width would be helpful. The manuscript is very strong without this figure, and given the complex feedback loops, it is hard to know if patterned expression drives growth or is a readout of growth.

Reviewer #3:

Tokamov et al. have used the *Drosophila* wing disc to explore mechanisms regulating of the abundance of the Hippo pathway upstream regulator, Kibra. They demonstrate that the increase in Kibra protein levels observed upon loss of Hippo pathway activity cannot be fully explained by the known Yki-dependent transcriptional feedback mechanism, and uncover a previously unsuspected post-translational layer of regulation of Kibra protein levels. They show that Kibra is phosphorylated and ubiquitylated in S2 cells, but less so upon knockdown of hpo and wts. They show that knockdown of the ubiquitin ligase substrate recognition subunit Slmb stabilises Kib, and identify a putative phospho-degron for Slmb within Kibra. They further show that mutation of this phospho-degron diminishes Kib ubiquitylation and renders Kibra insensitive to loss of Hippo pathway activity. They then use deletion constructs to narrow down the domains of Kibra necessary for degradation by their new mechanism and find that the regions essential for Kibra to activate the Hippo pathway are the ones necessary for degradation via their mechanism. Finally, they describe a non-uniform pattern of degradation of their Kibra stability reporters across the wing pouch.

The paper is well written and easy to read. The data are of high quality and the conclusions drawn are supported by the evidence presented. The findings are interesting, novel and should be of interest to the cell signalling and growth control fields. I support publication in *eLife*, provided the authors can clarify the points below.

Experimental points:

1) The authors make a good case that Kibra can be phosphorylated in a Hpo/Wts-dependent manner (even if it may not be directly by Hpo or Wts themselves), and that Hpo and Wts are required for Kibra stability. However, they don't directly test whether Wts kinase activity itself is required for Kibra degradation by Slimb. As they suggest that the assembly of a complex involving Kibra and several pathway members is necessary for degradation, it would be important to test if Wts kinase activity is required, or rather if it is the formation of this complex itself that is necessary to recruit Slimb. For example, this can be done by re-expressing wild type or kinase dead Wts in the S2 cell setup of Figure 3E (dsRNAs targeting the UTRs of wts would have to be used), and/or by looking at the effect of wild type or kinase dead Wts transgenes on ubi-kib-GFP expression in vivo.

2) To directly connect the Kibra degron identified by the authors with Hippo pathway activity, it would be good to look at expression of a Yki target gene such as ex-lacZ upon Kib-wt and Kib-S677A expression.

Text/presentation changes:

1) "These results suggest that the upstream regulation of the Hippo pathway is highly compartmentalized and that the activity of the core kinases is tightly associated with its distinct upstream regulators in a manner that allows little crosstalk between them".

Discussion – "Such tightly compartmentalized activity of the Hippo pathway is also suggested by the punctate appearance of all of these proteins at the cell cortex".

The authors should clarify in discussion what they mean with these statements. Is it that different microdomains of upstream regulators define hotspots of core kinase cascade activity that have different functions? What would this mean functionally for Yki inactivation?

2) The subsection “The Hippo pathway promotes Kibra degradation in a highly compartmentalized manner and independently of pathway activation by Expanded” refers to Figure 5F-G, but it should read 5D-E.

3) In Figure 6A (schematic of Kib truncation constructs), construct 1-857 has been drawn without a GFP tag at the end. I assume this is an omission.

4) The discussion about tension in the wing pouch (Discussion) is confusing. Surely if tension at the periphery of the pouch is higher, leading to decreased Hpo/Wts activity, then Kibra degradation should be lower there, not higher as observed? It therefore doesn't follow that "The results from this study may lead to the elucidation of how external cues, such as mechanical tension, modulate Kib-mediated Hippo signaling and pattern tissue growth." The authors should clarify this, or remove this part.

5) The authors should include scale bars in their images.

6) It would be nice to know which of their images are of live wing discs, and which are fixed (the appearance of reporters such as these can be different in live vs. fixed tissue). Could they list this in the Materials and methods?

---

## [Author Response]

Reviewer #1:In this manuscript, the authors present data to further unravel the complex regulation of the Hippo pathway, an important pathway controlling growth. Their data point to a negative feedback loop mediated by the core Hippo kinase cassette, which regulates one of its upstream regulators, Kibra, in a Yorkie-independent way. In addition, they provide some evidence that Kibra degradation is patterned across the disc. These are novel data and provide additional information on how such an important pathway is controlled to prevent excess growth of tissues, at least in some regions of the fly wing disc.1) They show that knock-down of Sav, Hippo or Warts results in increased Ubi>Kib-GFP levels, but I do not understand why they also "…… regulate its localization at the junctional cortex.". In both control and sav-, hpo- and wts-RNAi Ubi>Kib-GFP can be found at the junction (Supplementary Figure 1F).

We agree with the reviewer that this point is confusing to the reader. Because it is not important for the main focus of this manuscript, we have eliminated this statement from the text.

2) Figure 2I. The authors show dramatic decrease of ubiquitinated Ubi>Kib-GFP upon hpo- and wts-RNAi, as well as less phosphorylated Ubi>Kib-GFP upon single knock-down. No explanation is given why in one experiment they used single knock down (H), but double knock-down in (I). In addition, the statement that Kib levels are regulated "via phosphorylation-dependent ubiquitination" is an assumption, although it may be correct given findings published previously.

We initially used double knock-downs in this experiment to eliminate as much kinase activity as possible. We have since repeated these experiments using single knock-downs, which similarly affect Kibra ubiquitination. Figure 2I has been replaced with single knock-down experiments to avoid confusion over this point. Additionally, we agree that the notion that ubiquitination is phosphorylation dependent has not been shown directly and so have modified the text accordingly.

3) Subsection “Slimb regulates Kibra levels via a consensus degron motif”: If loss of Slmb leads to increased Ex (Ribeiro et al) and loss of either Slmb or Ex leads to increased Ubi>Kib-GFP (Figure 3A and Figure 1A, respectively), why should co-depletion of Ex and Slimb "… suppress the increase in Ubi>Kib-GFP levels”?

We thank the reviewer for pointing out that we were not sufficiently clear here. Our point was that because Kibra and Expanded have previously been shown to interact, our concern in this experiment was that the increased Kibra abundance observed upon depletion of Slimb could be due to increased recruitment by Expanded (whose abundance is greatly increased by Slimb depletion). The co-depletion experiment simply shows that increased Kibra is not caused by increased recruitment by Expanded. We have modified the text to make this point clearer to the reader.

We should also point out that the reviewer was mistaken in that Figure 1 shows endogenous Kibra (GFP-tagged via Crispr) rather than Ubi>Kib-GFP. In this case, we interpret the relatively modest increase in Kibra abundance to the previously identified feedback loop in which Yorkie promotes Kibra transcription. Ubi>Kib-GFP was used in subsequent experiments to eliminate the possibility of Yorkie-mediated transcriptional feedback.

4) Given their conclusion suggesting that the Kibra pathway is patterned, I wonder whether the authors always have studied the same region of the disc when comparing different genotypes. This is not always obvious from the figures.

With the exception of Figure 2, which includes images from the eye imaginal disc, all images are from the blade primordium (pouch) of the wing imaginal disc. We have added a statement in the Results indicating this. The main figures that involve wing imaginal discs show images of the entire wing pouch together with higher magnification insets which are indicated by boxes on the lower magnification panels (boxes have been added to Figure 6C-D). This allows the reader to see exactly what part of the wing blade is shown. As is apparent from the figures, we do not observe significant differences across the wing pouch in the effects we have described.

Reviewer #2:This interesting, well-written manuscript addresses the post-transcriptional regulation of the Hippo pathway regulator Kibra. The authors convincingly show that Kibra stability is regulated by Slimb, and that Hippo pathway kinases Hpo and Wts promote Kibra phosphorylation and thus Slimb-dependent degradation. They have strong biochemical and imaging based analyses. They define a function for a Slimb degron, and conduct structure function to explore Kibra domains in protein turnover. Overall this is an interesting, well conducted and novel story that is appropriate for publication in eLife.Endogenous Kib ( tagged with GFP) abundance is increased in merlin clones, but not in expanded clones, even though ex clones dramatically increase nuclear yorkie and ban3>gfp, and merlin does not. These data indicate loss of ex has a more potent effect on Yki than loss of merlin, and suggest Merlin acts independent of Yki and transcription. Consistent with this model, loss of Sd does not affect Kib levels in the eye. Importantly, ubiquitin driven KibraGFP is increased by loss of Merlin, Sav, Hpo or Wts. The authors note strong junctional accumulation of Kibra upon loss of Hpo or Wts. The authors show Kibra is phosphorylated and ubiquitinated upon Hpo/Wts activation. They show that Kibra has a Slimb degron (DSGVFE), and that mutation of this motif leads to stabilization of Kibra, and makes its levels insensitive to Hpo pathway activation. They also conducted structure function analyses that indicate that the WW domains of Kibra are important for its degradation upon Hpo pathway activation. Because Slimb promotes Ex degradation, loss of Slimb leads to increased Ex, and hence increased Hpo/Wts activity. The authors show overexpressed Ex does not in itself stabilize Kibra. While it is clear that Kibra phosphorylation is promoted by Hpo/Wts, it is still unclear if this is a direct phosphorylation The authors propose a model in which Kibra promotes Hpo/Wts activity, which then locally promotes Kibra degradation. It is not clear how it this local signaling is isolated from other junctional Hpo and Wts.

First, we thank the reviewer for these supportive comments. With regards to the final point, how this effect can be localized to just medial, Kibra-dependent signaling, we have added new findings to the revised version of the manuscript that we believe are relevant to this issue. As reviewer 3 noted, the data in the original version of the manuscript did not distinguish between a structural vs. enzymatic role for the pathway kinases Hippo and Warts. While the manuscript was being reviewed, we addressed this question using dsRNAs to deplete endogenously expressed Hippo or Warts in S2 cells and expressing RNAi-resistant, kinase dead mutant forms of either kinase. Interestingly, we found that the kinase dead forms of either kinase promote Kibra ubiquitination in S2 cells, indicating that it is formation of the signaling complex (presumably including Merlin, Kibra, Salvador, Hippo and Warts), rather than activation of the kinases, that recruits the Slimb E3 complex that ubiquitinates Kibra. We think this discovery nicely explains how this feedback mechanism is isolated from junctional signaling mediated by Expanded because it indicates that it is formation of the complex itself rather than activation of the kinases that promotes Kibra ubiquitination and degradation.

The weakest part of this very strong manuscript is Figure 7. If the intent is to show that Kibra is in a pattern in the wing, it would be good to include the staining and intensity quantification of endogenous Kibra in A and B comparison.

We thank the reviewer for this comment because it points out that we were not sufficiently clear in explaining the experiments presented in Figure 7. A key point that we now realize may not have been clear was that in these experiments Kibra, either wild-type or the degron mutant, was expressed ectopically under Nubbin>Gal4, rather than under its endogenous promoter. We did this because we did not want endogenous transcriptional patterning or transcriptional feedback from Yorkie to confound differences in protein degradation. So, although endogenous Kibra displays a similar abundance pattern to ectopically expressed wild-type Kibra (now shown in Figure 7—figure supplement 1A), we do not think that the endogenous pattern is directly relevant to the question we are asking here. Rather, we think the relevant comparison is between wild-type and degron mutant Kibra, where both forms are expressed identically at the transcriptional level and in a manner not subject to pathway feedback. As the figure shows, in this system the pattern of wild-type and degron mutant Kibra abundance across the wing blade is quite different, suggesting that Kibra degradation is patterned (non-uniform) across the blade. We note that this pattern is strikingly similar to the pattern of observed junctional tension that has been reported previously, leading us to propose that tension might regulate Kibra degradation.

Additionally, while the original version of the manuscript was under review we used a different experimental approach, designed in the Irvine lab, to further test the hypothesis that tension regulates Kibra abundance. This system uses expression of the microRNA *bantam* to produce small clones of rapidly growing cells in the wing imaginal disc. Because these clones are bounded by slower growing cells, they become compressed and cytoskeletal tension decreases (Pan et al., 2016). We set up this system in the background of either wild-type or ΔWW1 mutant Kibra-GFP expressed under the Ubi promoter. Like the degron mutant, ΔWW1 is not subject to pathway promoted degradation (because it is unable to assemble the complex), but unlike the degron mutant this transgene is not lethal under the Ubi promoter. While we see a robust increase in Kibra-GFP (wild-type) abundance in compressed *bantam*-expressing clones, we do not see any increase in the ΔWW1 mutant Kibra protein.

Taken together, we feel these two lines of experimental evidence provide evidence that the feedback regulation of Kibra we have identified here is, at least in part, regulated by mechanical tension. While we acknowledge that a fuller understanding of this will depend on understanding its underlying mechanistic basis, we think these interesting observations provide a developmental context that emphasizes the significance of the feedback we have discovered and therefore improve the impact of the work.

In Figure 7C and D, the authors should include pictures of examples of adult wings of the different genotypes that they provide the size analysis. Details of the landmarks on the wings they used to define distal and proximal width would be helpful.

Agreed – this has been done.

The manuscript is very strong without this figure, and given the complex feedback loops, it is hard to know if patterned expression drives growth or is a readout of growth.

Please see the comments just prior, and also those that follow in response to reviewer 3.

Reviewer #3:Tokamov et al. have used the *Drosophila* wing disc to explore mechanisms regulating of the abundance of the Hippo pathway upstream regulator, Kibra. They demonstrate that the increase in Kibra protein levels observed upon loss of Hippo pathway activity cannot be fully explained by the known Yki-dependent transcriptional feedback mechanism, and uncover a previously unsuspected post-translational layer of regulation of Kibra protein levels. They show that Kibra is phosphorylated and ubiquitylated in S2 cells, but less so upon knockdown of hpo and wts. They show that knockdown of the ubiquitin ligase substrate recognition subunit Slmb stabilises Kib, and identify a putative phospho-degron for Slmb within Kibra. They further show that mutation of this phospho-degron diminishes Kib ubiquitylation and renders Kibra insensitive to loss of Hippo pathway activity. They then use deletion constructs to narrow down the domains of Kibra necessary for degradation by their new mechanism and find that the regions essential for Kibra to activate the Hippo pathway are the ones necessary for degradation via their mechanism. Finally, they describe a non-uniform pattern of degradation of their Kibra stability reporters across the wing pouch.The paper is well written and easy to read. The data are of high quality and the conclusions drawn are supported by the evidence presented. The findings are interesting, novel and should be of interest to the cell signalling and growth control fields. I support publication in eLife, provided the authors can clarify the points below.Experimental points:1) The authors make a good case that Kibra can be phosphorylated in a Hpo/Wts-dependent manner (even if it may not be directly by Hpo or Wts themselves), and that Hpo and Wts are required for Kibra stability. However, they don't directly test whether Wts kinase activity itself is required for Kibra degradation by Slimb. As they suggest that the assembly of a complex involving Kibra and several pathway members is necessary for degradation, it would be important to test if Wts kinase activity is required, or rather if it is the formation of this complex itself that is necessary to recruit Slimb. For example, this can be done by re-expressing wild type or kinase dead Wts in the S2 cell setup of Figure 3E (dsRNAs targeting the UTRs of wts would have to be used), and/or by looking at the effect of wild type or kinase dead Wts transgenes on ubi-kib-GFP expression in vivo.

We completely agree with the reviewer's points here. In fact, while the original version of this manuscript was under review, we performed exactly the experiment requested. The results convincingly show that the kinase dead forms of Hippo and Warts promote Kibra ubiquitination quite effectively and thus argue that the kinases play a structural rather than enzymatic role in this process. This finding is quite exciting for two reasons. First, it potentially explains the remarkable compartmentalization between Kibra vs. Expanded promoted pathway activation because it argues that the physical presence of the kinases in a complex with Kibra, rather than their enzymatic activity (which might readily spread from one compartment to another), promotes Kibra degradation. Second, this finding suggests that another kinase is required to phosphorylate the degron sequence, a question we hope to solve in the future.

2) To directly connect the Kibra degron identified by the authors with Hippo pathway activity, it would be good to look at expression of a Yki target gene such as ex-lacZ upon Kib-wt and Kib-S677A expression.

While we agree in principle, in practice the difference in growth is subtle and these reporters are generally less sensitive to gain of upstream pathway function than its loss. Accordingly, we have added a statement in the text indicating that while the decreased growth caused by Kib-S677A is consistent with decrease Yki function, we have not formally shown this.

Text/presentation changes:1) "These results suggest that the upstream regulation of the Hippo pathway is highly compartmentalized and that the activity of the core kinases is tightly associated with its distinct upstream regulators in a manner that allows little crosstalk between them".Discussion – "Such tightly compartmentalized activity of the Hippo pathway is also suggested by the punctate appearance of all of these proteins at the cell cortex".The authors should clarify in the Discussion what they mean with these statements. Is it that different microdomains of upstream regulators define hotspots of core kinase cascade activity that have different functions? What would this mean functionally for Yki inactivation?

We thank the reviewer for these useful comments. We think the new data described above, showing that the kinases have a structural role in promoting Kibra degradation, substantially address the first point. That is, because it is the physical presence of the kinases in the complex, rather than their activity, the basis of the compartmentalization of this effect becomes clear. We have added text in the Discussion to clarify this point.

We agree that the second point is not really clear and since it is not a primary focus of this manuscript we have deleted it from the Discussion.

2) The subsection “The Hippo pathway promotes Kibra degradation in a highly compartmentalized manner and independently of pathway activation by Expanded” refers to Figure 5F-G, but it should read 5D-E.

Agreed.

3) In Figure 6A (schematic of Kib truncation constructs), construct 1-857 has been drawn without a GFP tag at the end. I assume this is an omission.

Yes, thank you for catching this mistake.

4) The discussion about tension in the wing pouch (Discussion) is confusing. Surely if tension at the periphery of the pouch is higher, leading to decreased Hpo/Wts activity, then Kibra degradation should be lower there, not higher as observed? It therefore doesn't follow that "The results from this study may lead to the elucidation of how external cues, such as mechanical tension, modulate Kib-mediated Hippo signaling and pattern tissue growth." The authors should clarify this, or remove this part.

Our new finding (described in detail above) that formation of the Kibra signaling complex, rather than activation of Hippo and Warts, promotes Kibra ubiquitination and degradation provides an elegant answer to this important question. We envision that as tension increases the degradation mechanism we have identified becomes more efficient (through mechanisms that remain to be discovered), resulting in degradation of Kibra and dissolution of the complex it assembles before it can efficiently activate the kinases and inactivate Yorkie. We have modified the Discussion to clarify this point.

We note that we also have added additional data, described above, showing that experimentally altering tension in the wing epithelium alters Kibra abundance.

5) The authors should include scale bars in their images.

We have done this.

6) It would be nice to know which of their images are of live wing discs, and which are fixed (the appearance of reporters such as these can be different in live vs. fixed tissue). Could they list this in the Materials and methods?

Agreed. This information has been added in the Materials and methods section. In short, all imaginal disc images except those of Expanded, Flag, and Scalloped antibody staining were done using live tissues.